# Dynamic repression by BCL6 controls the genome-wide liver response to fasting and steatosis

**Meredith A Sommars[1], Krithika Ramachandran[1], Madhavi D Senagolage[1], Christopher R Futtner[1], Derrik M Germain[1], Amanda L Allred[1], Yasuhiro Omura[1], Ilya R Bederman[2], Grant D Barish[1,3,4]\***

[1]Division of Endocrinology, Metabolism, and Molecular Medicine, Department of Medicine, Feinberg School of Medicine, Northwestern University, Chicago, United States; [2]Department of Pediatrics, Case Western Reserve University, Cleveland, United States; [3]Robert H. Lurie Comprehensive Cancer Center, Northwestern University, Chicago, United States; [4]Jesse Brown VA Medical Center, Chicago, United States

**Abstract** Transcription is tightly regulated to maintain energy homeostasis during periods of feeding or fasting, but the molecular factors that control these alternating gene programs are incompletely understood. Here, we find that the B cell lymphoma 6 (BCL6) repressor is enriched in the fed state and converges genome-wide with PPARα to potently suppress the induction of fasting transcription. Deletion of hepatocyte *Bcl6* enhances lipid catabolism and ameliorates high-fat-diet-induced steatosis. In *Ppara*-null mice, hepatocyte *Bcl6* ablation restores enhancer activity at PPARα-dependent genes and overcomes defective fasting-induced fatty acid oxidation and lipid accumulation. Together, these findings identify BCL6 as a negative regulator of oxidative metabolism and reveal that alternating recruitment of repressive and activating transcription factors to shared cis-regulatory regions dictates hepatic lipid handling.
DOI: https://doi.org/10.7554/eLife.43922.001

**\*For correspondence:**
grant.barish@northwestern.edu

**Competing interests:** The authors declare that no competing interests exist.

## Introduction

The classical studies of Jacob and Monod on the bacterial *lac* operon established a central paradigm for transcriptional repression to direct metabolic responses and sustain life in an environment of discontinuous food supply (*Jacob and Monod, 1961*; *Payankaulam et al., 2010*). In metazoans, nutrient-elicited transcription likewise coordinates the feeding to fasting transition of metabolism, yet a gap remains in our knowledge of the participating factors and their genomic coordination. In the fed state, sterol and carbohydrate regulatory element-binding proteins (SREBP and ChREBP) direct lipogenesis and glycolysis (*Abdul-Wahed et al., 2017*; *Horton et al., 2002*). Conversely, fasting disinhibits forkhead box transcription factors (FOXOs) and activates glucocorticoid receptor (GR) and cAMP response element binding protein (CREB) to promote gluconeogenesis (*Rui, 2014*). Extended fasting further stimulates peroxisome proliferator-activated receptor alpha (PPARα) to induce fatty acid oxidation, ketogenesis, and the fasting hormone FGF21 (*Badman et al., 2007*; *Inagaki et al., 2007*; *Kersten et al., 1999*; *Leone et al., 1999*). Despite progress revealing these various transcriptional activators, their dynamic genome-wide regulation and the influence of additional factors, particularly repressors, on the feeding to fasting transition remains poorly understood (*Goldstein and Hager, 2015*).

Recently, fasting-regulated enhancers were mapped using H3K27 acetylation ChIP- and DNase I hypersensitivity sequencing and footprinting, which inferred the presence of unknown repressors at

**eLife digest** Obesity has nearly tripled worldwide since the 1970s. A major health concern related to obesity is that excess fat can spill into organs such as the liver. This can lead to fatty liver disease or even liver cancer. Therefore, it is important to fully understand the mechanisms that lead to fat accumulation in the liver in order to develop new treatments.

Our bodies are designed to even out the highs and lows of an unpredictable diet by storing and releasing calories. When we are well-fed, liver cells switch on genes involved in making fat. When we have not eaten for a while, they switch them off and turn on genes involved in burning fat. Each switch involves thousands of genes, controlled by proteins called transcription factors. Some work as activators, turning genes on, whilst others work as repressors, turning genes off.

For example, the transcription factor PPAR alpha is a well-known activator that helps to regulate fat burning. However, we know much less about the repressors that stop cells burning fat when there is plenty of food available. To find out more, Sommars et al. studied the repressor BCL6 in mouse liver cells.

The results revealed that BCL6 interacts with hundreds of the same genes as PPAR alpha. When the mice were eating, BCL6 turns off the genes involved in fat burning, but when they were starved PPAR alpha activated those genes. However, when BCL6 was experimentally removed, many fat-burning genes were permanently switched on. So, even when mice were fed a high-fat diet, they burned off fat in their livers.

Understanding the role of genetic switches like PPAR alpha and BCL6 is crucial for understanding how and why our bodies store energy. This could help us to create treatments that enhance the liver's ability to burn excess fat.

DOI: https://doi.org/10.7554/eLife.43922.002

regions enriched with STAT motifs (*Goldstein et al., 2017*). Our focus turned to B-cell lymphoma 6 (BCL6), a key immune cell repressor with affinity for STAT-like DNA recognition sequences (*Dent et al., 1998*; *Dent et al., 1997*; *Zhang et al., 2012*). BCL6 is a member of the ZBTB family of C2H2-type zinc finger proteins and represses transcription through a variety of interactions with cor-epressors including SMRT, NCoR, BCoR, CtBP, MTA3/NuRD, and HDACs (*Basso and Dalla-Favera, 2012*). Although well-recognized for critical roles in B-cell and T-cell development and lymphoma-genesis, BCL6 is also broadly expressed outside of the immune system where its functions are largely unknown.

In this work, using genome-wide DNA binding and transcriptomic analyses as well as hepatocyte-specific gene targeting, we reveal an unexpected role for BCL6 as a potent antagonist of PPARα-directed gene regulation. We find that BCL6 and PPARα bind independently at thousands of shared regulatory regions in sub-nucleosomal proximity, often at multiple locations along the same gene. Genes harboring these BCL6-PPARα regulatory modules constitute over 50% of fasting-responsive transcripts and exhibit particularly dynamic expression. Moreover, we find that ablation of hepato-cyte *Bcl6* increases lipid oxidation, prevents high-fat-diet-induced steatosis, and reverses fasting-related defects in *Ppara*$^{-/-}$ mice including aberrant enhancer activity, transcription, ketosis, and lipid accumulation. These restorations in *Ppara*$^{-/-}$ mice devoid of liver *Bcl6* were linked to loss of HDAC3-containing BCL6 repressive complexes and enhanced recruitment of PPARδ to BCL6-PPAR shared enhancers. Together, these findings establish BCL6 as a critical repressor of oxidative metabolism.

## Results

### BCL6 colocalizes with PPARα at fasting-regulated genes controlling lipid oxidation

To establish the genomic sites for BCL6 regulation, we used ChIP-seq to map its genome-wide set of cis-acting targets (cistrome) in liver. Under fed conditions, we identified over fifteen thousand high confidence BCL6 binding sites from three biological replicates. Ontologies for nearby genes were dominated by lipid and ketone metabolism, PPAR signaling, and functions in peroxisomes and mitochondria (*Figure 1A*). Additionally, motif analysis of BCL6 binding sites compared to random

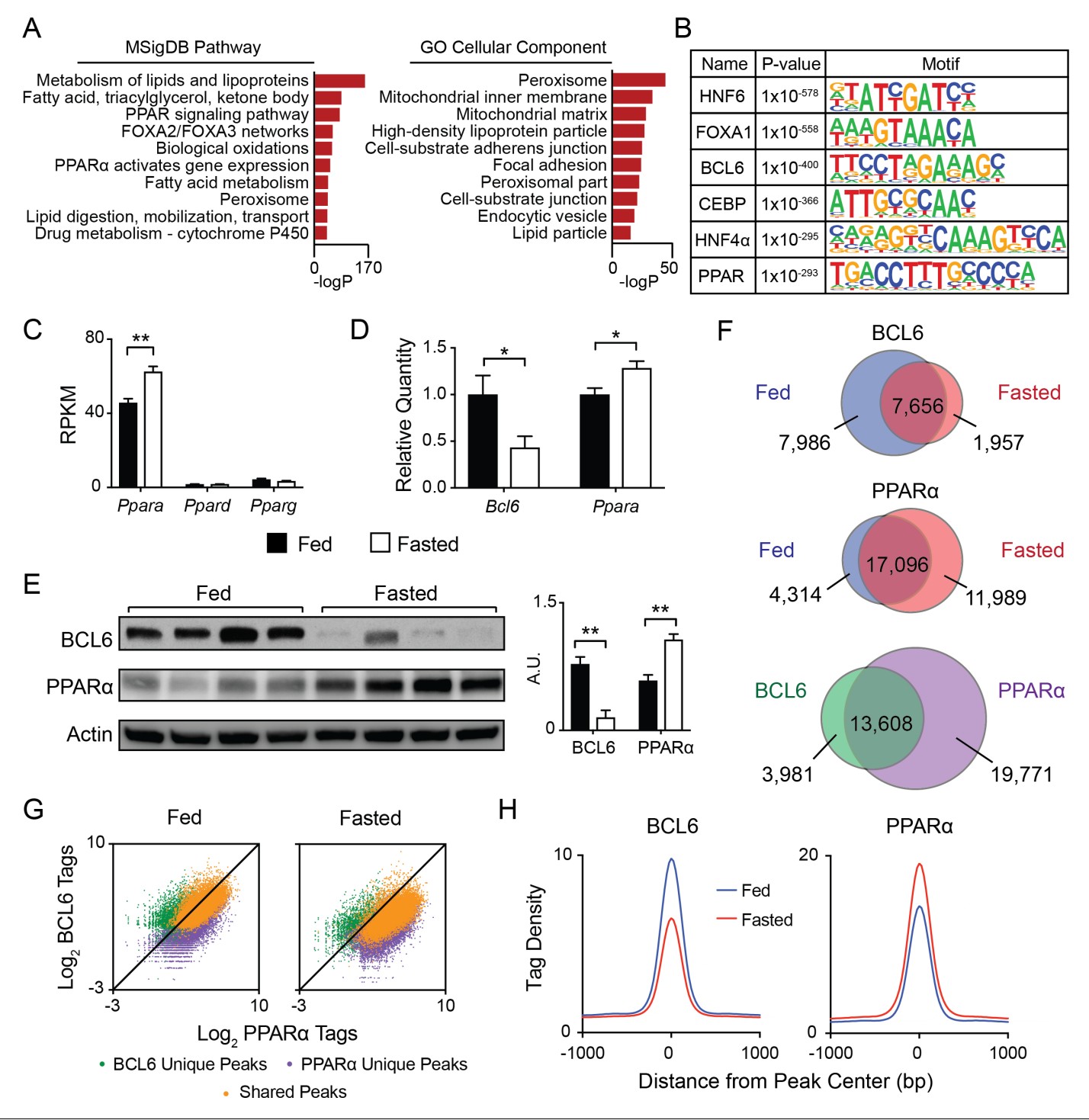

**Figure 1.** BCL6 converges with PPARα at fasting-regulated lipid genes. (A) MSigDB Pathway and Gene Ontology (GO) Cellular Compartment analysis of BCL6 ChIP-seq binding sites. (B) Motif enrichment analysis of BCL6-bound regions. (C) Gene expression as measured by reads per kilobase of transcript per million reads (RPKM) of *Ppara*, *Ppard*, and *Pparg* in control (*Bcl6*<sup>fl/fl</sup>) mouse liver samples. N = 4 per group. (D) qPCR of *Bcl6* and *Ppara* in fed and fasted *Bcl6*<sup>fl/fl</sup> mouse livers. N = 6–7 per group. (E) Western blot analysis of BCL6 and PPARα protein levels in *ad libitum* fed and overnight fasted C57BL/6 mouse livers. Densitometry normalized to actin levels is shown (right). (F) Venn diagrams comparing liver ChIP-seq peaks from *ad libitum* and overnight fasted mice using antibodies against BCL6 (top) and PPARα (middle). Overlap between combined fed and fasted BCL6 and PPARα binding sites (based on a distance between peak centers of <200 bp) is shown (bottom). ChIPs were performed in biological triplicates. (G) BCL6 and PPARα ChIP-seq tag densities at BCL6 unique, PPARα unique, or shared BCL6-PPARα peaks in fed and fasted states. (H) BCL6 (left) and

*Figure 1 continued on next page*

*Figure 1 continued*

PPARα (right) tag densities at BCL6-PPARα shared peaks in fed or fasted livers. N = 3 per group. A two-tailed Student's t-test assuming equal variance was used to compare mean values between two groups. Data are represented as mean ±SEM. *p<0.05, **p<0.01, ***p<0.001.

DOI: https://doi.org/10.7554/eLife.43922.003

The following figure supplements are available for figure 1:

**Figure supplement 1.** BCL6 liver ChIP-seq reflects binding events specific to hepatocytes.

DOI: https://doi.org/10.7554/eLife.43922.004

**Figure supplement 2.** BCL6 and PPARα reciprocally bind to shared regulatory regions near fasting genes.

DOI: https://doi.org/10.7554/eLife.43922.005

whole genome sequences revealed striking enrichment of response elements not only for BCL6 but also for lipid-activated PPAR nuclear hormone receptors (*Figure 1B*) (*Evans et al., 2004*), the pioneer factor FOXA1, the enhancer remodeler C/EBP (*Grøntved et al., 2013*), and the developmental and lipid regulatory factors HNF4 (*Hayhurst et al., 2001*; *Li et al., 2000*) and HNF6 (*Clotman et al., 2005*; *Zhang et al., 2016*). Highly similar BCL6 peak calling, gene ontology and motif analysis was obtained using either wild-type liver input chromatin or BCL6 ChIP-seq from livers of hepatocyte-specific *Bcl6* knockouts (*Bcl6*[LKO] mice) as background controls for enrichment (*Figure 1—figure supplement 1A and B*) indicating that the liver BCL6 cistrome reflected binding events specific to hepatocytes.

Based on motif predictions, we pursued the possibility of genomic convergence between BCL6 and PPARs. Direct quantification of TF consensus sites near BCL6 binding sites further reflected enrichment of motifs for PPARs, its heterodimeric partner RXR, and to a lesser extent FXR, whereas motifs for other abundant liver transcription factors such as LXR were absent (*Figure 1—figure supplement 1B*). *Ppara* is the dominantly expressed PPAR subtype in liver (*Figure 1C*). In line with PPARα's critical role to regulate the adaptive response to fasting, its RNA and protein levels increase with overnight food deprivation (*Figure 1C–E*) (*Kersten et al., 1999*). In contrast, *Bcl6* mRNA and corresponding protein diminish sharply from the fed to the fasted state (*Figure 1D and E*). Accordingly, BCL6 occupancy was diminished at the majority of its binding sites and its cistrome was reduced by 39%, whereas PPARα recruitment was enhanced and its cistrome was expanded by 36% with fasting (*Figure 1F*, top and middle panels, and *Figure 1—figure supplement 2A*, left panel). In addition, fasting resulted in a redistribution of binding sites for each factor. Direct comparison of the combined fed and fasted ChIP-seq peaks for BCL6 and PPARα revealed 13,608 overlapping binding regions (<200 bp between peak centers) between these factors, representing 77% and 41% of the BCL6 and PPARα cistromes, respectively (*Figure 1F*, bottom panel). Of these overlapping peaks, the vast majority (>96%) demonstrated a distance of <100 bp between peak centers (*Figure 1—figure supplement 2B*). Over 95% of these overlapping sites occurred outside of promoter regions in intragenic and intergenic locations (*Figure 1—figure supplement 2C*). BCL6-PPARα co-occurring peaks represented the strongest binding events for each factor, indicating they likely represent true DNA interactions as opposed to non-specific events at open chromatin regions (*Figure 1G*) (*Landt et al., 2012*). At these shared sites, binding by BCL6 decreased while PPARα increased upon fasting (*Figure 1H* and *Figure 1—figure supplement 2A*, right panel), which was evident at several PPARα target genes, such as *Acot4/3* and *Por* (*Figure 1—figure supplement 2D and E*) and confirmed by ChIP qPCR (*Figure 1—figure supplement 2F*). Thus, extensive cistromic overlap and reciprocal genome-wide binding suggested BCL6 and PPARα may control a common regulatory program.

## Genomic localization of BCL6 is independent of PPARα and PPARδ

Next, we assessed whether BCL6 and PPARs compete or collaborate for DNA binding. Using livers from *Ppara*[-/-] and wild-type control mice, we found that ablation of *Ppara* had no impact on BCL6 enrichment at BCL6-PPARα binding sites (*Figure 2A*, left panel and *Figure 1—figure supplement 2D*). Likewise, liver-specific deletion of *Bcl6* did not alter PPARα binding (*Figure 2A*, right panel and *Figure 1—figure supplement 2E*). *Ppard* is expressed at relatively low levels in liver (*Figure 1C*), but it was previously reported that unliganded PPARδ binds and sequesters BCL6, releasing it in the presence of PPARδ ligands (*Lee et al., 2003*). Thus, to test whether a protein complex between

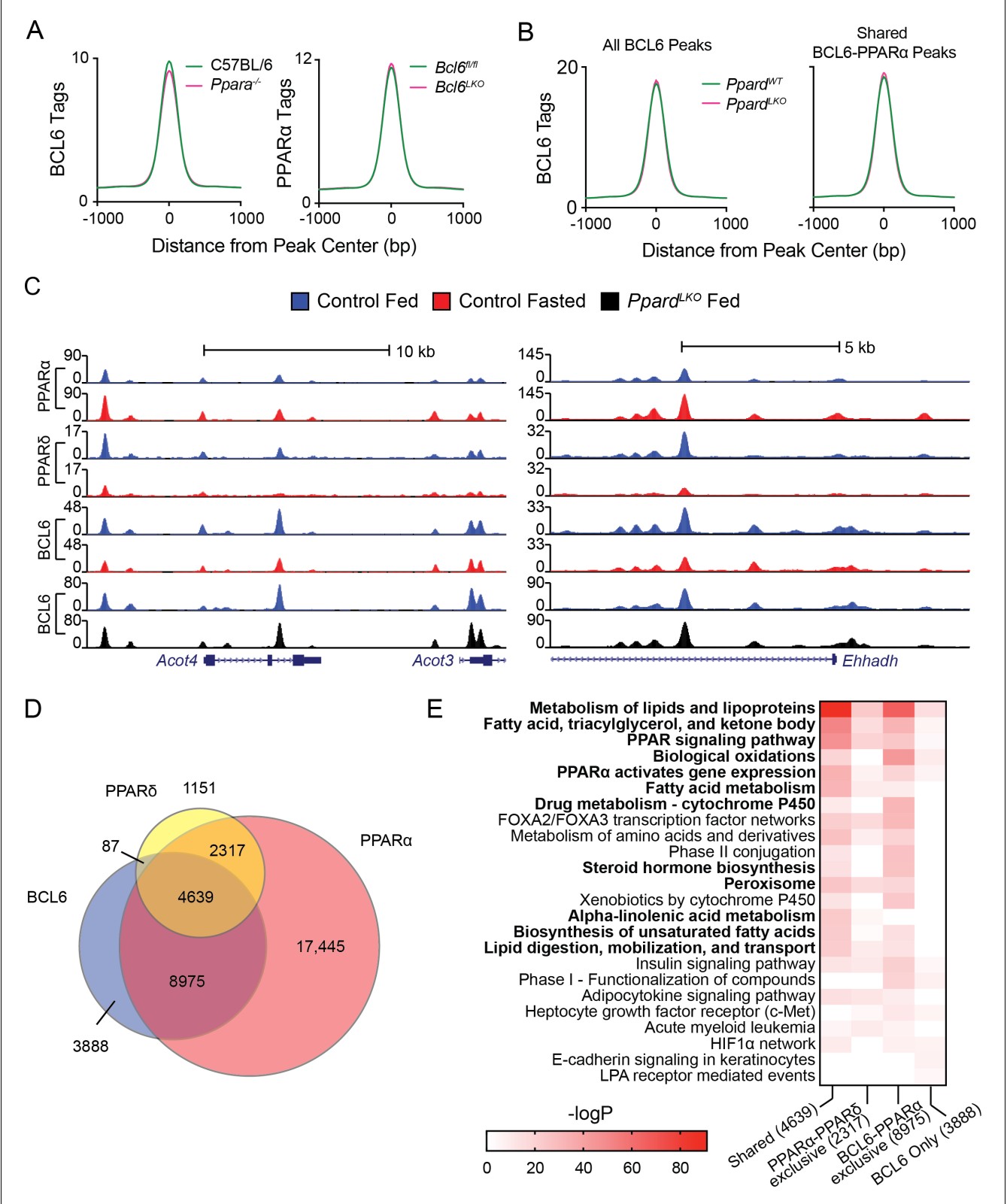

**Figure 2.** BCL6 genome-wide DNA binding is independent of PPARα and PPARδ. (**A**) BCL6 ChIP-seq tag densities in C57BL/6 and *Ppara⁻/⁻* livers at shared BCL6-PPARα peaks (left). PPARα tag density in *Bcl6^fl/fl* and *Bcl6^LKO* livers at shared BCL6-PPARα peaks (right). (**B**) BCL6 tag densities in control and *Ppard^LKO* mouse livers at all BCL6 peaks (left) or shared BCL6-PPARα peaks (right), N = 2 per group. (**C**) UCSC genome browser tracks showing PPARα, PPARδ, and BCL6 ChIP-seq in control fed livers (blue), control fasted livers (red), or *Ppard^LKO* livers (black). (**D**) Venn diagram showing overlap of

*Figure 2 continued on next page*

*Figure 2 continued*

PPARδ, BCL6, and PPARα cistromes in mouse liver. Cistromes for each factor include peaks identified in either fed or fasted livers. Peaks were considered overlapping if peak centers were within 200 bp. ChIPs were performed in biological triplicates. (**E**) Gene ontology enrichment for binding regions common among BCL6, PPARα, and PPARδ (Shared); exclusive to PPARα and PPARδ (PPARα-PPARδ); exclusive to BCL6 and PPARα (BCL6-PPARα); or exclusive to BCL6 (BCL6 only).

DOI: https://doi.org/10.7554/eLife.43922.006

The following figure supplement is available for figure 2:

**Figure supplement 1.** PPARα and PPARδ compete for binding at shared sites.

DOI: https://doi.org/10.7554/eLife.43922.007

PPARδ and BCL6 could account for BCL6-PPAR genomic co-localization, we characterized BCL6 binding in the presence or absence of hepatocyte PPARδ using mice harboring floxed alleles of *Ppard* and Albumin-Cre (*Ppard*[LKO] mice). The livers of *Ppard*[LKO] animals exhibited 96% decreased levels of *Ppard* mRNA with no significant change in *Bcl6* (*Figure 2—figure supplement 1A*), yet in comparison to wild type control livers, BCL6 binding was unaltered across the BCL6 cistrome and at its subset of BCL6-PPARα shared sites (*Figure 2B and C*). Thus, these findings did not support a model in which BCL6 binds to PPARs on chromatin.

Additionally, we mapped the liver PPARδ cistrome using an isotype-specific antibody. 8,194 PPARδ-binding sites were identified collectively in fed and fasted livers, 85% of which overlapped with the more extensive PPARα cistrome of 33,379 sites (*Figure 2D*). Overall, PPARδ binding was diminished by half upon fasting (*Figure 2—figure supplement 1B*), but this reduction was only evident at sites shared with PPARα such as the *Acot4/3* and *Ehhadh* loci (*Figure 2C*), suggesting that PPARα and PPARδ compete for binding at common response elements (*Figure 2—figure supplement 1C*). While PPARδ and BCL6 co-localized at only 87 genomic sites without PPARα, we detected 8,975 BCL6-PPARα peaks which were not bound by PPARδ (*Figure 2D*). Gene ontology analysis revealed that BCL6-PPARα-PPARδ shared or BCL6-PPARα exclusive peaks annotate predominantly to genes controlling the metabolism of lipids and lipoproteins, fatty acids, triacylglycerol, ketone bodies, PPAR signaling, and biological oxidations (*Figure 2E*). Collectively, these results provided further evidence that extensive BCL6 genome-wide colocalization with PPARα and, to a more limited degree, with PPARδ occurs due to independent, yet proximate DNA-binding events along genes controlling lipid metabolism.

## SMRT/NCoR-HDAC3 complexes control acetylation in hepatocyte BCL6-bound regulatory regions

To better understand how BCL6 modulates gene expression in liver, we first identified the BCL6-regulated transcriptome. We generated mice with hepatocyte-specific *Bcl6* deletion (*Bcl6*[LKO]) by crossing animals with floxed alleles of *Bcl6* to mice expressing *Cre* under control of the albumin enhancer/promoter. *Bcl6*[LKO] mice exhibited 75% reduced *Bcl6* mRNA and over 90% diminished protein levels in the liver (*Figure 3—figure supplement 1A and B*). In *ad lib* fed *Bcl6*[LKO] mice, RNA-seq revealed 721 upregulated genes, while only 362 were downregulated by more than two fold compared to controls (*Figure 3A*). These findings indicated that liver BCL6 predominantly functions as a repressor of transcription, which was particularly apparent at genes with strongly bound BCL6-binding sites (*Figure 3B*).

BCL6 is known to control transcription in immune cells through interactions with many different cofactors (*Barish et al., 2012*; *Basso and Dalla-Favera, 2012*; *Hatzi et al., 2013*). To test whether BCL6 regulates transcription through similar interactions in liver, we used ChIP-seq to characterize SMRT, NCoR, and HDAC3 binding in *ad lib* fed *Bcl6*[fl/fl] and *Bcl6*[LKO] mice. In *Bcl6*[fl/fl] animals, we found extensive cistrome overlap between BCL6 and all three corepressors (6,643 common sites), although 21% of BCL6 sites were unique (*Figure 3C*). SMRT and HDAC3 exhibited very few independent binding regions, with only ~2% unique for either cofactor. NCoR exhibited the most extensive cistrome, and 45% of its sites did not overlap with BCL6, HDAC3, or SMRT. In line with their known biochemical interactions, SMRT, NCoR, and HDAC3 peaks were enriched in motifs for nuclear receptors (ERR, PPAR, RXR) as well as FOX and HNF transcription factors when compared to whole genome DNA as background (*Figure 3—figure supplement 2A*) (*Perissi et al., 2010*).

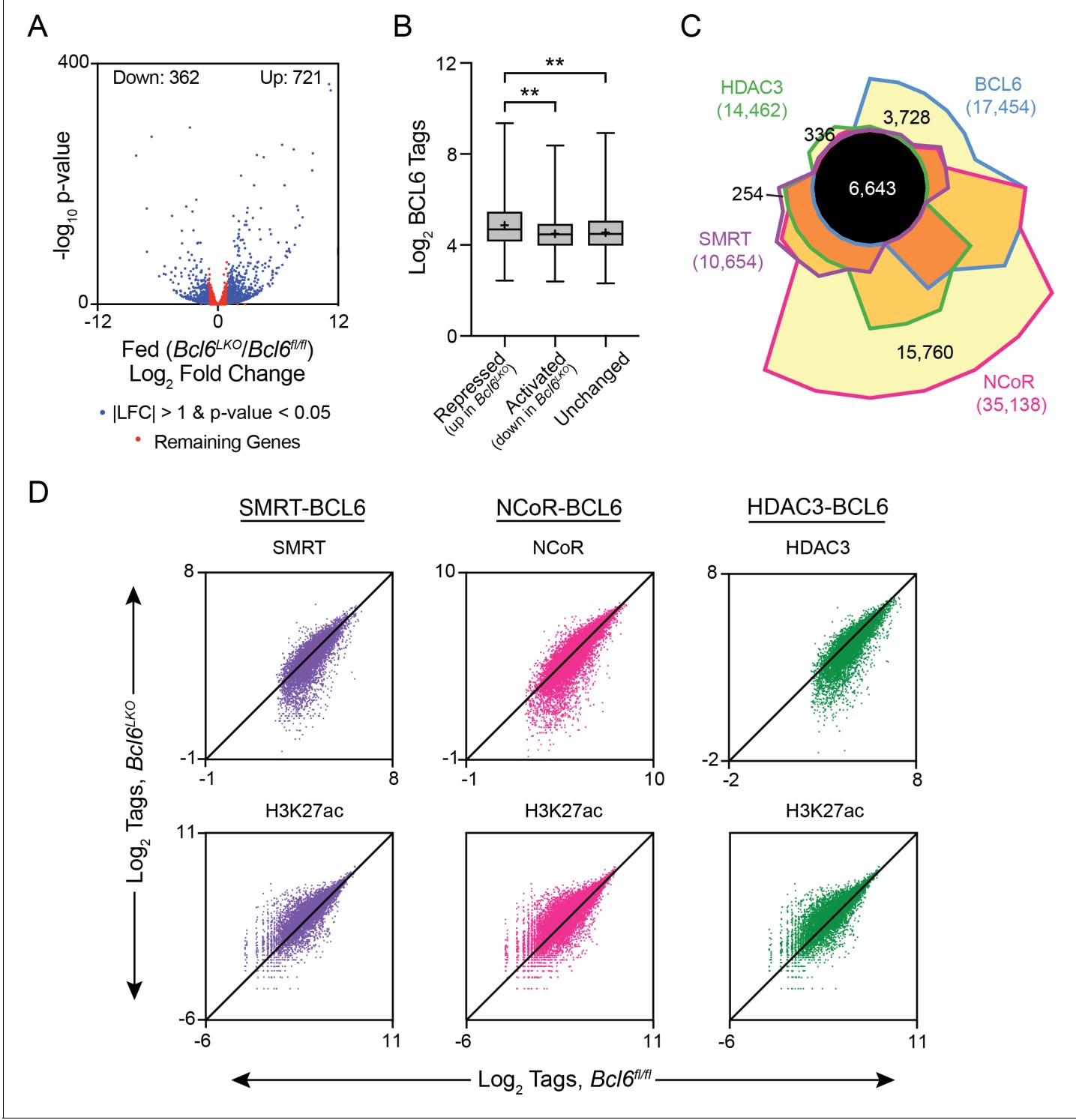

**Figure 3.** BCL6 complexes with SMRT/NCoR-HDAC3 to reduce H3K27ac and represses transcription. (A) Volcano plots showing log$_2$ fold change (LFC) in expression of fed $Bcl6^{LKO}$ over $Bcl6^{fl/fl}$ livers. Blue dots represent |LFC| greater than one with an adjusted p-value less than 0.05. Red dots represent remaining expressed genes. N = 4 per group. (B) BCL6 tags in control livers at BCL6 peaks near BCL6-activated, -repressed, or -unchanged genes. Box plots display interquartile range (box), median (horizontal black line), mean (black '+'), and min to max (whiskers). (C) Four-way Venn diagram comparing $ad\ lib$ fed control BCL6, SMRT, NCoR and HDAC3 ChIP-seq peak sets. (D) Tag density of H3K27ac, SMRT, NCoR, and HDAC3 ChIP-seq in $Bcl6^{fl/fl}$ and $Bcl6^{LKO}$ livers at respective cofactor peaks co-bound with BCL6. ChIPs were performed in biological triplicates. For (B), a one-way ANOVA and Tukey's post-hoc testing was used to compare tag density between groups. *p<1×10$^{-3}$, **p<1×10$^{-6}$, ***p<1×10$^{-9}$, ****p<1×10$^{-12}$.

*Figure 3 continued on next page*

*Figure 3 continued*

DOI: https://doi.org/10.7554/eLife.43922.008

The following figure supplements are available for figure 3:

**Figure supplement 1.** *Bcl6^LKO^* mice exhibit efficient protein and mRNA reductions of BCL6.

DOI: https://doi.org/10.7554/eLife.43922.009

**Figure supplement 2.** SMRT, NCoR, and HDAC3 are recruited to chromatin by BCL6 in liver.

DOI: https://doi.org/10.7554/eLife.43922.010

For each corepressor, we further analyzed peaks shared with BCL6 (peak centers colocalizing within 200 bp) and non-overlapping (unique) cofactor sites (*Figure 3—figure supplement 2A*). When compared against DNA sequences from unique peaks, shared peaks were overrepresented with motifs for BCL6, STAT, and FOX transcription factors, as well as CUX2 and HNF6. In contrast, when tested against the DNA sequences of BCL6-shared peaks, unique SMRT, NCoR, and HDAC3 sites were enriched in motifs for ETS and ELK transcription factors. Next, we quantified SMRT, NCoR, and HDAC3 occupancy at BCL6-binding sites that colocalized with corepressor peaks in control versus *Bcl6^LKO^* livers (*Figure 3D* and *Figure 3—figure supplement 2B*). For each corepressor, binding at BCL6 sites was significantly reduced in *Bcl6^LKO^* livers. Moreover, loss of these complexes was inversely correlated to histone 3 lysine 27 acetylation (H3K27ac), a marker for enhancer activity (*Creyghton et al., 2010*; *Wang et al., 2008*), which was significantly elevated along BCL6-SMRT/NCoR-HDAC3 sites in *Bcl6^LKO^* livers. Together, these findings revealed a role for BCL6 to recruit a subset of liver SMRT/NCoR-HDAC3 complexes and repress associated regulatory regions.

## Ablation of Bcl6 de-represses a fasting gene program

Gene ontology analysis of differentially expressed transcripts in the livers of *Bcl6^LKO^* animals revealed lipid metabolism, oxidation, and PPAR signaling as top scoring terms (*Figure 4A*). This regulatory signature and the extensive genomic intersection between BCL6 and PPARα prompted us to determine whether BCL6 could likewise control fasting-induced gene expression. Livers from mice restricted from food overnight exhibited 162 genes upregulated and 174 genes downregulated by at least 2-fold using RNA-seq (*Figure 4B*), and fasting regulated a common set of gene expression pathways with *Bcl6* ablation (*Figure 4A*). Notably, over 40% of robustly regulated fasting genes (135/336) were controlled by BCL6 (*Figure 4C*, top panel) and for the vast majority, *Bcl6* ablation mimicked the impact of fasting on transcription (*Figure 4C*, bottom panel and *Figure 4D*). Unsupervised clustering analyses of liver gene expression revealed that patterns in *Bcl6^LKO^* mice, irrespective of nutrition status, more closely resembled profiles from fasting than fed control mice (*Figure 4D* and *Figure 4—figure supplement 1A*). Genes co-regulated by fasting and *Bcl6* deletion are enriched in ontologies for lipid and ketone body metabolism as well as PPARα signaling (*Figure 4—figure supplement 1B*). For example, visualization of ChIP-seq and RNA-seq tracks demonstrated that PPARα and BCL6 reciprocally occupy regions along the *Acot4/3* and *Vnn1* genes, whose expression was strongly induced by either fasting or *Bcl6* ablation (*Figure 4E*). Quantitative PCR further confirmed dozens of liver genes that were similarly upregulated by fasting or *Bcl6* ablation, including many involved in mitochondrial and peroxisomal β-oxidation (*Abcd1/2, Acadvl, Acnat2, Acot2, Acot3/4, Ehhadh, Hadh, Idh2, Ucp2*), microsomal ω-hydroxylation (*Aldh3a2, Cyp4a31*), ketogenesis (*Acss3, Bdh1, Fgf21, Hmgcl*), and lipid metabolism (*Abhd2, Acot1, Cd36*) (*Figure 4—figure supplement 1C*). Together, these results suggested that loss of *Bcl6* mimics the fasting-induced transcriptional program controlling liver lipid metabolism.

## BCL6-PPARα regulatory regions cluster on dynamically transcribed fasting genes

We next examined the extent to which BCL6 and PPARα cis-regulatory sites alone or in combination control fasting transcription. Hypergeometric testing revealed a 1.1-fold enrichment (p-value 3.8e-14) for BCL6-PPARα peaks relative to the entirety of PPARα genome-wide peaks along all genes differentially regulated (p-value<0.05) by fasting. Over 50% of these fasting genes contained co-occurring BCL6-PPARα-binding sites (*Figure 4—figure supplement 2A*), with a median of two co-occurring sites per gene (*Figure 4—figure supplement 2B*). By contrast, just 24% or 1.4% of fasting

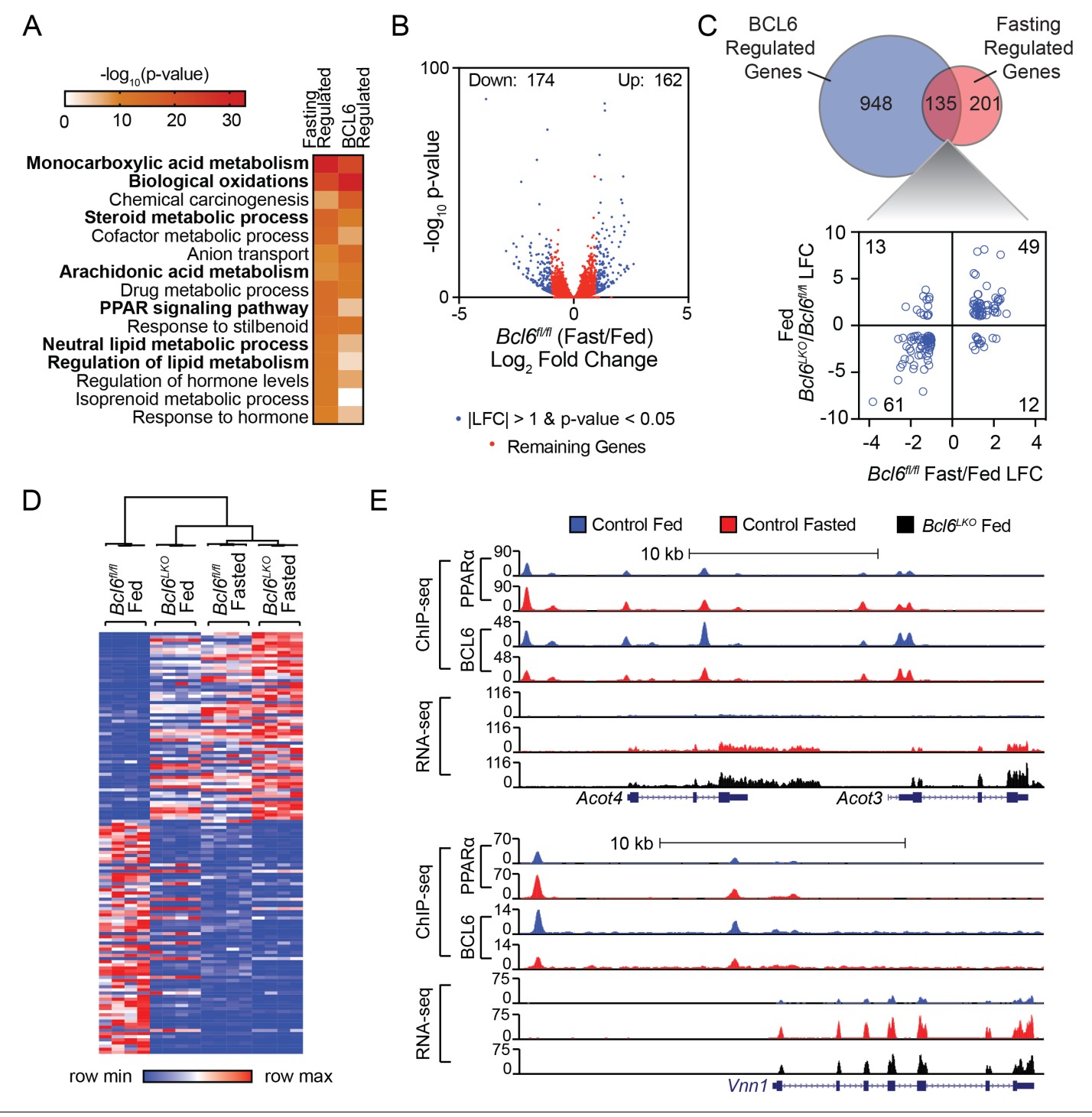

**Figure 4.** *Bcl6* deletion de-represses a fasting liver gene program. (A) Pathway enrichment analysis for genes differentially expressed (|LFC| greater than one and an adjusted p-value less than 0.05) with fasting or *Bcl6* deletion in liver. (B) Volcano plots showing log₂ fold change (LFC) in expression of *Bcl6^fl/fl* fasted over fed. Blue dots represent |LFC| greater than one with an adjusted p-value less than 0.05. Red dots represent remaining expressed genes. N = 4 per group. (C) Venn diagram comparing differentially expressed genes with fasting or *Bcl6* deletion (top panel). Comparison of LFC between fed (*Bcl6^LKO*/*Bcl6^fl/fl*) (y-axis) and *Bcl6^fl/fl* (fasted/fed) (x-axis) for genes differentially expressed by both fasting and *Bcl6* deletion is shown (bottom panel). (D) Hierarchical clustering heatmap of RPKM values in fed and fasted *Bcl6^fl/fl* and *Bcl6^LKO* samples for genes regulated by both fasting and *Bcl6* deletion. N = 4 per group. (E) UCSC genome browser tracks of BCL6 and PPARα ChIP-seq and RNA-seq data at PPARα-regulated genes, *Acot4/3* and *Vnn1* in control fed (blue), control fasted (red), and *Bcl6^LKO* fed (black) livers.

DOI: https://doi.org/10.7554/eLife.43922.011

*Figure 4 continued on next page*

Figure 4 continued

The following figure supplements are available for figure 4:

**Figure supplement 1.** BCL6 deletion mimics the fasting gene program.

DOI: https://doi.org/10.7554/eLife.43922.012

**Figure supplement 2.** The BCL6-PPARα regulatory module clusters along dynamic fasting genes.

DOI: https://doi.org/10.7554/eLife.43922.013

genes contained PPARα-only or BCL6-only sites, respectively, and these occurred with a median of just one regulatory region per gene. In addition, fasting-regulated genes with BCL6-PPARα regulatory elements exhibited significantly greater ranges of expression than those with PPARα peaks that lack this heterotypic module (*Figure 4—figure supplement 2C*), and their ontology was particularly enriched for functions in lipid regulation and oxidative metabolism (*Figure 4—figure supplement 2D*). In summary, over half of fasting-regulated genes are controlled by BCL6-PPARα-binding sites, and this gene subset is particularly dynamic in transcription.

## Liver Bcl6 ablation restores fasting expression and enhancer activity in *Ppara*$^{-/-}$ mice

PPARα is critical for the fasting induction of genes mediating peroxisomal and mitochondrial fatty acid β-oxidation as well as microsomal ω-hydroxylation (*Contreras et al., 2013*; *Gao et al., 2015*; *Hardwick et al., 2009*; *Hashimoto et al., 2000*; *Kersten et al., 1999*; *Leone et al., 1999*; *Montagner et al., 2016*). To determine whether loss of the BCL6 repressor in liver compensates for transcriptional defects in *Ppara*$^{-/-}$ mice, we generated animals with combined whole body deletion of *Ppara* and liver-specific ablation of *Bcl6* (*Ppara*$^{-/-}$;*Bcl6*$^{LKO}$ mice). RNA-seq revealed that loss of *Bcl6* rescued 209 of 795 dysregulated genes in fasted *Ppara*$^{-/-}$ mice compared to fasted controls (*Figure 5A*). Among genes normally upregulated with fasting, *Bcl6* deletion restored expression of genes involved in monocarboxylic acid and lipoprotein metabolism; ketone body synthesis; AMPK and PPAR signaling; and peroxisomes (*Figure 5B*, top panel). By contrast, genes normally downregulated upon fasting and rescued in *Ppara*$^{-/-}$;*Bcl6*$^{LKO}$ mice represented pathways mostly unrelated to lipid metabolism (*Figure 5B*, bottom panel). Restoration of *Ppara*$^{-/-}$ defective fasting transcription in *Ppara*$^{-/-}$;*Bcl6*$^{LKO}$ mice was confirmed by qPCR at genes involved in β-oxidation (*Acot2/3/4*, *Idh2*), ω-hydroxylation (*Aldh3a2*, *Cyp4a31*), ketone body synthesis (*Acss3*, *Fgf21*, *Hmgcl*, *Hmgcs2*), and lipid metabolism (*Abhd2*, *Cd36*, *Vldlr*) (*Figure 5—figure supplement 1*). Thus, loss of *Bcl6* restores expression at a subset of PPARα-directed genes controlling lipid metabolism.

Opposing regulation between PPARα and BCL6 was also observed at the level of chromatin modification. We profiled histone H3K27ac in overnight fasted *Bcl6*$^{fl/fl}$ control, *Ppara*$^{-/-}$, and *Ppara*$^{-/-}$;*Bcl6*$^{LKO}$ mice using ChIP-seq (*Figure 5C*, left panel). Fasting-induced genes with impaired expression in *Ppara*$^{-/-}$ mice demonstrated low H3K27ac signal around BCL6-PPARα-binding sites in *Ppara*$^{-/-}$ compared to control mice. By contrast, in livers of *Ppara*$^{-/-}$;*Bcl6*$^{LKO}$ animals, H3K27ac is reestablished or even enhanced at these sites (*Figure 5C*, top left panel; and *Figure 5—figure supplement 2A*). Reciprocal H3K27ac patterns were found at impaired fasting-repressed genes in *Ppara*$^{-/-}$ and *Ppara*$^{-/-}$;*Bcl6*$^{LKO}$ animals (*Figure 5C*, bottom left panel). This pattern in H3K27ac at BCL6-PPARα sites occurred only at fasting impaired genes that were rescued in *Ppara*$^{-/-}$;*Bcl6*$^{LKO}$ mice (*Figure 5—figure supplement 2B*). Thus, BCL6 de-repression restores aberrant liver cis-regulatory activity in *Ppara*$^{-/-}$ mice along fasting responsive genes.

## Loss of liver *Bcl6* relieves HDAC3-associated repression and enhances recruitment of PPARδ to shared BCL6-PPARα sites

Next, we sought to further understand how ablation of hepatocyte *Bcl6* could rescue fasting expression defects in *Ppara*$^{-/-}$ mice. The reestablishment of acetylation at fasting enhancers with BCL6-PPARα sites pointed to a shift in the balance of transcription factor complexes with histone deacetylase (HDAC) and acetyltransferase (HAT) activities at these co-regulated regions. Since hepatocyte BCL6 binds to SMRT/NCoR-HDAC3 at a subset of its binding sites (*Figure 3C and D*), we specifically examined HDAC3 occupancy at BCL6-PPARα peaks along rescued fasting genes. In the absence of *Bcl6*, HDAC3 was substantially diminished at these BCL6-PPARα sites (*Figure 5D*). Additionally, we

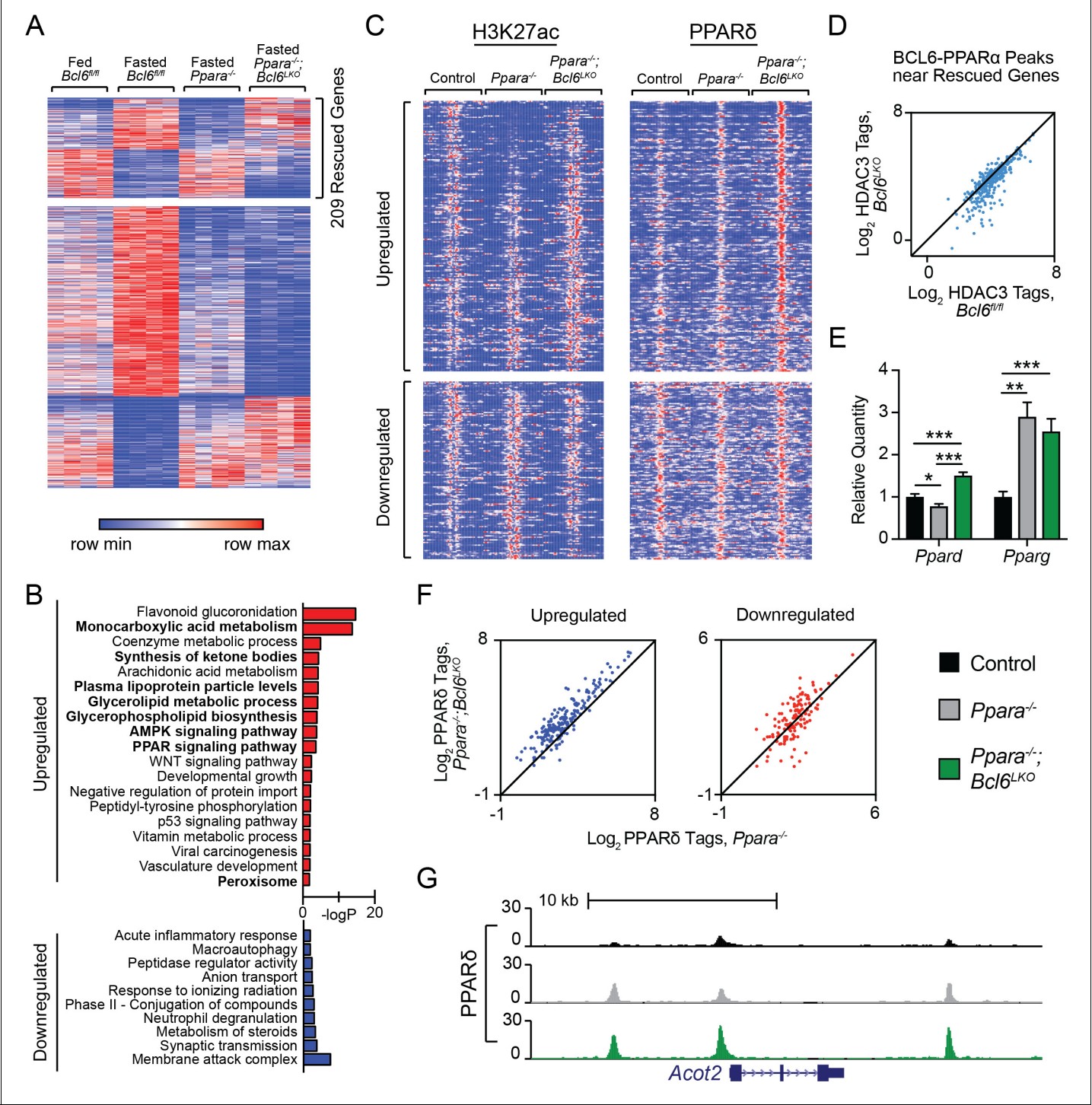

**Figure 5.** *Bcl6* ablation reduces HDAC3 activity and promotes PPARδ binding in *Ppara*[−/−] mice at rescued genes. (**A**) Heatmap of RPKM values at fasting-regulated genes that are dysregulated in *Ppara*[−/−] mice. 209 genes have partially or completely restored expression upon fasting in *Ppara*[−/−]; *Bcl6*[LKO] mice. N = 4 per group. (**B**) Gene ontologies of upregulated (top) and downregulated (bottom) fasting genes restored in *Ppara*[−/−];*Bcl6*[LKO] mice. (**C**) Heatmap of H3K27ac and PPARδ ChIP-seq in fasted control *Bcl6*[fl/fl], *Ppara*[−/−], and *Ppara*[−/−];*Bcl6*[LKO] mice at BCL6-PPARα shared peaks that annotate to rescued *Ppara*[−/−] dysregulated genes in *Ppara*[−/−];*Bcl6*[LKO] mice. N = 3 per group. (**D**) HDAC3 ChIP-seq tag density in *Bcl6*[fl/fl] and *Bcl6*[LKO] livers at BCL6-PPARα shared peaks near rescued fasting genes. (**E**) qPCR of *Ppard* and *Pparg* in fasted control, *Ppara*[−/−], and *Ppara*[−/−];*Bcl6*[LKO] mice. N = 5–6 per group. (**F**) PPARδ tag density at rescued upregulated (left) and downregulated (right) fasting genes in fasted *Ppara*[−/−];*Bcl6*[LKO] and *Ppara*[−/−] livers. ChIP was performed in biological triplicate. (**G**) UCSC genome browser track of PPARδ ChIP-seq at *Acot2*. In (**E**), a one-way ANOVA and Holm-Sidak's post-hoc testing was used to compare mean expression between groups. *p<0.05, **p<0.01, ***p<0.001.

*Figure 5 continued on next page*

*Figure 5 continued*

DOI: https://doi.org/10.7554/eLife.43922.014

The following figure supplements are available for figure 5:

**Figure supplement 1.** BCL6 deletion in *Ppara*[-/-] mice restores fasting gene expression.

DOI: https://doi.org/10.7554/eLife.43922.015

**Figure supplement 2.** *Bcl6* deletion is linked to enhanced PPARδ binding in fasted Pparα[-/-] mice.

DOI: https://doi.org/10.7554/eLife.43922.016

tested whether BCL6 could influence other PPAR isotypes, which can be associated with CBP/p300 HAT complexes that acetylate H3K27 (*Jin et al., 2011*). Using qPCR, we found that *Ppard* levels were significantly increased in fasted *Ppara*[-/-];*Bcl6*[LKO] compared to *Ppara*[-/-] mice, while *Pparg* levels were unchanged (*Figure 5E*). Moreover, BCL6 ChIP-sequencing revealed that BCL6 binds multiple intronic sites along the *Ppard* gene (*Figure 5—figure supplement 2C*), suggesting that it directly represses *Ppard* expression. Consistent with their enhanced *Ppard* levels, we observed increased PPARδ binding near rescued genes in *Ppara*[-/-];*Bcl6*[LKO] compared to *Ppara*[-/-] mice (*Figure 5C*, right panel), particularly at upregulated fasting genes (*Figure 5F*), including *Acot2*, (*Figure 5G*), *Hmgcs2*, *Aldh3a2* (*Figure 5—figure supplement 2D*), and others (*Figure 5—figure supplement 2E*). Together, these observations identified that loss of BCL6 directly relieves repression and potentiates PPARδ-mediated transactivation to restore fasting liver gene expression in *Ppara*[-/-];*Bcl6*[LKO] mice.

## *Bcl6* ablation enhances hepatic lipid catabolism and reduces steatosis

Next, we determined the functional impact of the BCL6 regulatory program on hepatic regulation and lipid processing *in vivo*. *Ad libitum* fed *Bcl6*[LKO] mice exhibited higher circulating ketone bodies compared to *Bcl6*[fl/fl] mice (*Figure 6A*). This difference persisted after a 24 hr fast. Additionally, mice lacking hepatic *Bcl6* have higher rates of complete fatty acid oxidation as measured by oxidation of $^{14}$C-palmitate in liver homogenates (*Figure 6B*). In contrast, analysis of fatty acid uptake, triglyceride secretion, and hepatic lipogenesis based on *in vivo* deuterium incorporation revealed no other differences in lipid metabolism between *Bcl6*[LKO] mice and controls (*Figure 6C–E*). To test a broader role for BCL6 in lipid processing, we assessed hepatic triglyceride content after feeding mice high-fat diet (HFD) for 19 weeks. *Bcl6*[LKO] mice were profoundly protected from developing steatosis, as demonstrated by oil red O staining and more than a 50% reduction in hepatic triglyceride content compared to *Bcl6*[fl/fl] controls, despite similar increases in body weight (*Figure 6F–H*). Accompanying these reductions in hepatic lipid accumulation, HFD-exposed *Bcl6*[LKO] mice exhibited significantly lower levels of fasting serum glucose (*Figure 6—figure supplement 1A*) and a non-significant reduction in insulin (*Figure 6—figure supplement 1B*). Moreover, when challenged with a shorter term 5-week HFD, *Bcl6*[LKO] mice exhibited a trend towards enhanced insulin responsiveness, as measured by levels of phosphorylated AKT following acute administration of exogenous insulin (*Figure 6—figure supplement 1C*). These combined observations demonstrate that mice lacking hepatic *Bcl6* have heightened capacity to catabolize lipids via β-oxidation and subsequent ketogenesis or TCA cycling, as well as improved glucose homeostasis when challenged with high-fat diet.

*Ppara*[-/-] mice exhibit fasting hypoketonemia and impaired fatty acid oxidation leading to steatosis (*Gao et al., 2015*; *Hashimoto et al., 2000*; *Kersten et al., 1999*; *Leone et al., 1999*; *Montagner et al., 2016*). After 48 hr of fasting, *Ppara*[-/-] mice developed centrilobular macrosteatosis (*Figure 6I*), as previously reported (*Hashimoto et al., 2000*). Remarkably, *Ppara*[-/-];*Bcl6*[LKO] animals were strongly protected from hepatic triglyceride accumulation based upon histological analysis with oil red O staining and demonstrated 23% reduced triglyceride accumulation compared to *Ppara*[-/-] mice (*Figure 6I and J*). Compared to fasted *Ppara*[-/-] mice, *Ppara*[-/-];*Bcl6*[LKO] mice also had higher rates of $^{14}$C-palmitate oxidation in liver homogenates, exhibited in both completely oxidized $^{14}$CO$_2$ and incompletely oxidized $^{14}$C-acid soluble intermediates (*Figure 6K*). In line with their reduced lipid accrual, *Ppara*[-/-];*Bcl6*[LKO] mice also revealed higher ketone body levels (*Figure 6L*), suggesting that ablation of *Bcl6* can de-repress ketone body synthesis even in the absence of *Ppara*. Overall, these results established that loss of liver *Bcl6* rescues metabolic defects of *Ppara* deficiency.

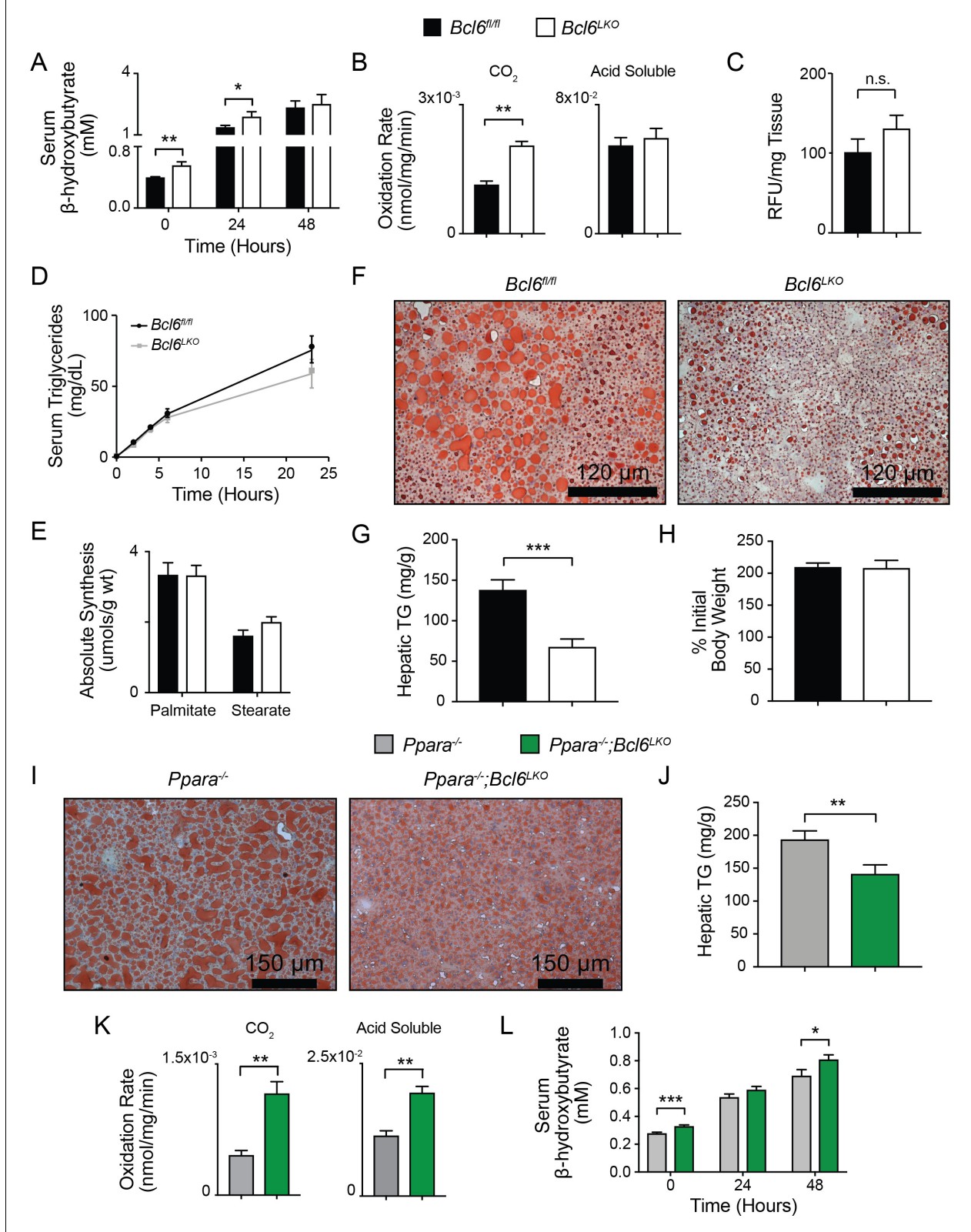

**Figure 6.** *Bcl6* deletion enhances fatty acid oxidation and ameliorates steatosis. (**A**) Serum β-hydroxybutyrate levels were measured in mice over the course of a 48 hr fast. N = 8–12 per group. (**B**) Rates of $^{14}C$-palmitate oxidation in *Bcl6$^{fl/fl}$* and *Bcl6$^{LKO}$* liver homogenates measured in $CO_2$ and acid soluble fractions. N = 3 per group. (**C**) *In vivo* lipid uptake quantified by bodipy $C_{16}$ assays in *Bcl6$^{fl/fl}$* and *Bcl6$^{LKO}$* mice. N = 7 per group. (**D**) Lipid secretion measured by serum triglyceride sampling over time after injecting *Bcl6$^{fl/fl}$* and *Bcl6$^{LKO}$* mice with Poloxomer. N = 11–13 per group. (**E**) *In vivo*

*Figure 6 continued on next page*

*Figure 6 continued*

palmitate and stearate synthesis determined by $^2$H incorporation in *Bcl6$^{fl/fl}$* and *Bcl6$^{LKO}$* livers. N = 8–10 per group. (F) Oil red O staining in livers, (G) biochemical quantification of liver triglycerides, and (H) % change in body weight in *Bcl6$^{fl/fl}$* and *Bcl6$^{LKO}$* mice following 19 weeks on 45% high fat diet. N = 7–11 per group. (I) Oil red O staining in livers and (J) biochemical quantification of liver triglycerides from *Ppara$^{-/-}$* and *Ppara$^{-/-}$;Bcl6$^{LKO}$* mice following a 48 hr fast. N = 16–18 per group. (K) Rates of $^{14}$C-palmitate oxidation in *Ppara$^{-/-}$* and *Ppara$^{-/-}$;Bcl6$^{LKO}$* liver homogenates measured in $CO_2$ and acid soluble fractions. N = 4–5 per group. (L) Serum β-hydroxybutyrate levels were measured in mice over the course of a 48 hr fast. N = 8–17 per group. A two-tailed Student's t-test assuming equal variance was used to compare means between two groups. Data are represented as mean ±SEM. *p<0.05, **p<0.01, ***p<0.001.

DOI: https://doi.org/10.7554/eLife.43922.017

The following figure supplement is available for figure 6:

**Figure supplement 1.** Mice lacking hepatic *Bcl6* have improved insulin sensitivity after high-fat diet.

DOI: https://doi.org/10.7554/eLife.43922.018

## Discussion

Dynamic transcriptional programming is necessary to sustain life in environments of varying access to food, and the liver is central to orchestrate systemic metabolism in response to such changes. However, our understanding of the epigenomic programs that underpin feeding and fasting metabolism is limited and dominated by studies of hormonally-cued transcriptional activators. Our work has identified BCL6 as a potent repressor of lipid catabolism, both in the context of fasting and dietary lipid overload. On a genome-wide scale, BCL6 converges with PPARα at over 13,000 regulatory regions on which BCL6 binding is enriched with feeding, whereas PPARα is induced by fasting. This dynamic BCL6-PPARα cis-regulatory module annotates to over 1,400 fasting-responsive genes. Moreover, *Bcl6* ablation mimics the fasting transcriptional response, and a myriad of defects in *Ppara$^{-/-}$* mice are partially rescued by concomitant deletion of hepatocyte *Bcl6*, ranging from defective fasting enhancer activity and gene expression to impaired fatty acid oxidation, hypoketonemia, and susceptibility to steatosis. Together, these findings evidence a powerful role for BCL6 to epigenomically oppose PPARα and to suppress fatty acid oxidation.

Previously, BCL6 functions outside of hematopoietic cells were poorly defined. In liver, prior analysis supported a role for BCL6 in competing with STAT5 and modulating responses to growth hormone and drug metabolism (*Chikada et al., 2018*; *Zhang et al., 2012*). Further, a study of whole-body knockout mice posited a role for BCL6 in systemic metabolism, but it was confounded by analysis limited to animals with severe and frequently fatal inflammatory disease (*LaPensee et al., 2014*). Original characterization of *Bcl6$^{-/-}$* mice demonstrated variable degrees of growth retardation and ill health within three weeks of life, with half dying before 5 weeks of age (*Dent et al., 1997*). Over 80% of *Bcl6$^{-/-}$* mice exhibit myocarditis and over 70% have pulmonary vasculitis with elevated levels of IL-4,–5, and −13, cytokines known to directly impact liver metabolism (*Ricardo-Gonzalez et al., 2010*; *Stanya et al., 2013*). Thus, metabolic phenotyping of whole body *Bcl6* knockout mice was uninformative (*LaPensee et al., 2014*), and the role for BCL6 in cell-intrinsic hepatic lipid metabolism was previously unknown. Using genetic, genomic, and isotopic analyses we reveal that loss of *Bcl6* in hepatocytes causes cell-autonomous enhancement of fatty acid oxidation without a direct impact on lipid synthesis.

Physical interactions between BCL6 and various cofactors have been well documented and in immune cells mediate distinct functional roles (*Huang et al., 2014*; *Huang et al., 2013*). Among these interaction partners are SMRT and NCoR, which bind to the BCL6 N-terminal BTB domain and function as scaffolds to recruit HDAC3 and other corepressive machinery (*Perissi et al., 2010*). In liver, we found nearly 80% overlap between BCL6 and the cistromes of SMRT, NCoR, and HDAC3. Moreover, loss of BCL6 was associated with significantly diminished occupancy of these cofactors at BCL6-bound regulatory regions (*Figure 3*). However, thousands of SMRT, HDAC3, and particularly NCoR binding peaks were independent of BCL6. Furthermore, even at regulatory regions where BCL6 and these coregulators colocalize, persistent ChIP-seq signals for SMRT, NCoR, and HDAC3 are often observed in the genetic absence of *Bcl6*. These findings indicate that SMRT, NCoR, and HDAC3 may frequently engage multiple transcription factor complexes within a single regulatory region. Given their extensive interactions, there is tremendous complexity in deciphering corepressor roles in metabolic regulation. Indeed, knockouts and various knockin mutants of NCoR, SMRT,

and HDAC3 have demonstrated hepatic steatosis phenotypes (*Knutson et al., 2008*; *Mottis et al., 2013*; *Shimizu et al., 2015*; *Sun et al., 2012*), in contrast to the lipid overload-protected phenotype observed here with hepatocyte *Bcl6* ablation. The cofactor requirements for BCL6-mediated control in the liver and extent to which SMRT/NCoR-HDAC3 are responsible for its potent repression of lipid catabolism warrant further investigation.

The clustering of transcription factors at regulatory regions has been proposed as a flexible mechanism to control diverse gene expression patterns during development and in response to environmental stimulus (*Arnone and Davidson, 1997*; *Smith et al., 2013*). Motif enrichment indicated a relationship between the BCL6 repressor and the PPAR subfamily of lipid-activated nuclear receptors. PPARα is the predominant PPAR isotype in liver, while PPARδ is expressed at lower levels but was reported to physically interact with BCL6 (*Lee et al., 2003*). However, we find that *Ppara* and *Ppard* are each genetically dispensable for chromatin recruitment of BCL6. Thus, BCL6 opposition to PPARα occurs via proximate binding at independent cis-regulatory elements, a mechanism distinct from FXR, which has been reported to counter PPARα transcriptional outputs through competition for DR1-binding sites (*Lee et al., 2014*). The regulatory interaction between BCL6 and PPARα is also unique from other integrative regulators of hepatic lipid metabolism such as HNF6 and REV-ERBα, which cooperatively repress transcription via tethering (*Zhang et al., 2016*). In addition to PPARs, it is possible that other transcriptional activators predicted to converge with BCL6 including HNF6 and HNF4, FOXA1, and C/EBP (*Figure 1B*), collaborate with BCL6 and PPARα in hepatic lipid regulation (*Hayhurst et al., 2001*; *Zhang et al., 2016*).

The BCL6-PPARα regulatory module is remarkable for its widespread occurrence along genes controlling lipid catabolism. We speculate that BCL6-PPAR elements in fasting enhancers endow them with variably repressive or activating regulatory potential, contributing to highly dynamic gene expression across the feeding to fasting transition. In a related manner, genetic ablation of *Bcl6* de-represses these regulatory regions and compensates for loss of PPARα transactivity in *Ppara*<sup>-/-</sup>; *Bcl6*<sup>LKO</sup> mice. Remarkably, this occurs both directly, via loss of active repression at BCL6-PPAR elements, and indirectly by upregulating *Ppard* to enhance transactivity at BCL6-PPAR sites. Thus, we find that liver metabolic shifts are not simply directed by inducible transactivating factors. Rather, 'active repression' (*Hanna-Rose and Hansen, 1996*) by BCL6 and its dynamic modulation play key additional roles in toggling between the fed and fasted state and determining hepatic lipid accumulation. Since inhibitors of BCL6 have been developed to target BCL6 and selective interactions with its corepressors (*Cardenas et al., 2016*; *Lu et al., 2018*), these findings also raise the possibility that BCL6 de-repression could represent a future therapeutic strategy for non-alcoholic fatty liver disease.

## Materials and methods

**Key resources table**

| Reagent type (species) or resource | Designation | Source or reference | Identifiers | Additional Information |
|---|---|---|---|---|
| Genetic reagent (*Mus musculus*) | *Bcl6*<sup>fl/fl</sup> | PMID 30566857 | | |
| Genetic reagent (*Mus musculus*) | *Albumin-cre* | Jackson Laboratory | Stock #003574 | |
| Genetic reagent (*Mus musculus*) | *Ppara*<sup>-/-</sup> | Jackson Laboratory | Stock #008154 | |
| Genetic reagent (*Mus musculus*) | *Ppard*<sup>fl/fl</sup> | Jackson Laboratory | Stock #005897 | |
| Antibody | anti-BCL6 (guinea pig polyclonal) | PMID 30566857 | | custom polyclonal 7.5 µg per IP |
| Antibody | anti-PPARδ (guinea pig polyclonal) | PMID 28467934 | | custom polyclonal 7.5 µg per IP |
| Antibody | anti-SMRT (guinea pig polyclonal) | PMID 22465074 | | custom polyclonal 7.5 µg per IP |

*Continued on next page*

Continued

| Reagent type (species) or resource | Designation | Source or reference | Identifiers | Additional Information |
|---|---|---|---|---|
| Antibody | anti-NCoR (guinea pig polyclonal) | PMID 22465074 | | custom polyclonal 7.5 µg per IP |
| Antibody | anti-HDAC3 (rabbit polyclonal) | Santa Cruz | Cat. #: sc-11417x | 5 µg per IP |
| Antibody | anti-H3K27ac (rabbit polyclonal) | Active Motif | Cat. #: 39133 | 5 µg per IP |
| Antibody | anti-pAKT (rabbit monoclonal) | Cell Signaling | Cat. #: 4060 s | (1:1000) |
| Antibody | anti-panAKT (rabbit monoclonal) | Cell Signaling | Cat. #: 4691 s | (1:1000) |
| Antibody | anti-BCL6 (mouse monoclonal) | Santa Cruz | Cat. #: sc7388 | (1:200) |
| Antibody | anti-β-Actin (mouse monoclonal) | Sigma-Aldrich | Cat. #: A1978 | 1:1000) |
| Antibody | anti-PPARα (rabbit polyclonal) | Santa Cruz | Cat. #: sc-9000x | 7.5 µg per IP; WB: (1:500) |
| Antibody | Peroxidase AffiniPure Goat Anti-Mouse IgG | Jackson Immuno Research | Cat. #: 115-035-174 | (1:20,000) |
| Antibody | Peroxidase IgG Fraction Monoclonal Mouse Anti-Rabbit IgG | Jackson Immuno Research | Cat. #: 211-032-171 | (1:20,000) |
| Antibody | Rabit Anti-guinea pig IgG H and L | Abcam | Cat. #: ab6698 | |
| Chemical compound, drug | DSG Crosslinker | ProteoChem | Cat. #: c1104 | |
| Chemical compound, drug | Formaldehyde, 16%, methanol-free, Ultra Pure | Polysciences, Inc | Cat. #: 18814–20 | |
| Chemical compound, drug | RNAlater Stabilization Solution | ThermoFisher Scientific | Cat. #: AM7020 | |
| Chemical compound, drug | TRIzol Reagent | ThermoFisher Scientific | Cat. #: 15596018 | |
| Chemical compound, drug | Poloxamer 407 | Sigma-Aldrich | Cat. #: 16758 | |
| Chemical compound, drug | cOmplete Ultra Tablets, EDTA-free | Sigma-Aldrich | Cat. #: 5892953001 | |
| Chemical compound, drug | Deuterium oxide | Sigma-Aldrich | Cat. #: 151882 | |
| Chemical compound, drug | Sodium Palmitate | Sigma-Aldrich | Cat. #: P9767 | |
| Chemical compound, drug | Palmitic Acid, [1–14C] | MP Biomedicals | Cat. #: 12195 | |
| Chemical compound, drug | BODIPY 500/510 C1, C12 | Invitrogen | Cat. #: D3823 | |
| Chemical compound, drug | Humulin R | Lilly | NDC 0002-8215-01 | |
| Commercial assay or kit | Dynabeads M-280 Sheep Anti-Rabbit IgG | Invitrogen | Cat. #: 11204D | |
| Commercial assay or kit | Dynabeads M-280 Tosylactivated | Invitrogen | Cat. #: 14204 | |
| Commercial assay or kit | Protein A Agarose/Salmon Sperm DNA | Millipore | Cat. #: 16–157 | |
| Commercial assay or kit | iScript cDNA Synthesis Kit | BioRad | Cat. #: 1708891 | |
| Commercial assay or kit | iTaq Universal SYBR Green | BioRad | Cat. #: 1725124 | |
| Commercial assay or kit | Infinity Triglyceride Assay Kit | ThermoFisher Scientific | Cat. #: TR22421 | |
| Commercial assay or kit | β-Hydroxybutyrate (Ketone Body) Colorimetric Assay Kit | Cayman Chemical | Cat. #: 700190 | |
| Commercial assay or kit | Ultra Sensitive Mouse Insulin ELISA Kit | Crystal Chem | Cat. #: 90080 | |

*Continued*

| Reagent type (species) or resource | Designation | Source or reference | Identifiers | Additional Information |
|---|---|---|---|---|
| Commercial assay or kit | Glucose Colorimetric /Fluorometric Assay Kit | BioVision | Cat. #: K606 | |
| Commercial assay or kit | Microvette CB 300 K2E | Sarstedt | Cat. #: 16.444 | |
| Commercial assay or kit | MemCode Reversible Protein Stain Kit | ThermoFisher Scientific | Cat. #: 24585 | |
| Commercial assay or kit | Whatman qualitative filter paper | Sigma-Aldrich | Cat. #: WHA1003055 | |
| Commercial assay or kit | KAPA Hyper Prep Library Prep Kit | Kapa Biosystems | Cat. #: KK8504 | |
| Commercial assay or kit | KAPA Stranded RNA-seq Kit with RiboErase | Kapa Biosystems | Cat. #: KK8483 | |
| Commercial assay or kit | NextSeq 500/550 High Output Kit v2.5 (75 cycles) | Illumina | Cat. #: 20024906 | |
| Other, research diet (45% kcal from fat) | HFD | Research Diets, Inc | Stock #D12451 | |

## Mice

*Bcl6*$^{fl/fl}$ mice were generated through the UC Davis Mouse Biology Program by engineering loxP sites between exons 5 and 6 of the mouse *Bcl6* locus. Cre-mediated deletion creates a frameshift mutation, resulting in a protein of 138 amino acids (compared to 708 amino acids in wild-type BCL6) lacking exons 5–10 and the zinc finger DNA binding domain. *Ppara*$^{-/-}$ (Stock #008154) and *Ppard*$^{fl/fl}$ (Stock #005897) mice were obtained from Jackson Laboratories. *Bcl6*$^{fl/fl}$ and *Ppard*$^{fl/fl}$ mice were crossed with *Albumin-Cre* animals (Jackson Laboratories, Stock #003574) to generate *Bcl6*$^{fl/fl}$; *Albumin-Cre* (*Bcl6*$^{LKO}$) and *Ppard*$^{fl/fl}$; *Albumin-Cre* (*Ppard*$^{LKO}$) mice, respectively. Mice were maintained on a 14:10 light: dark (LD) cycle with free access to water. Unless otherwise specified, 'fed' refers to *ad libitum* feeding with standard chow and 'fasted' refers to a 16–18 hr overnight fast. High-fat diet containing 45% of kcal from fat was obtained from Research Diets, Inc (Stock #D12451). All animal care and use procedures were conducted in accordance with regulations of the Institutional Animal Care and Use Committee at Northwestern University.

## Chromatin immunoprecipitation

Chromatin immunoprecipitation (ChIP) was performed as previously described (*Barish et al., 2010*). ChIP samples were prepared in biological triplicate (three animals per condition), unless otherwise specified. Mouse livers were harvested, rinsed in PBS, and crosslinked at room temperature for 30 min in 2 mM disuccinimidyl glutarate and then for 10 min in 1% formaldehyde. After quenching with 125 mM glycine, crosslinked material was rinsed twice with cold PBS and frozen at −80°C until further processing. Crosslinked material was lysed in buffer containing 0.75M NaCl, 1% Triton X, 0.5 mM Tris, 0.05 mM EDTA, and 0.5% NP-40. Isolated nuclei were then sheared in buffer containing 1% SDS, 10 mM EDTA, and 50 mM Tris for six cycles (30 s on, 30 s off) using a Diagenode Bioruptor to shear chromatin into 200–1000 bp fragments. Protein-DNA complexes were incubated overnight with antibody against BCL6 (custom polyclonal to mouse BCL6), PPARα (Santa Cruz), PPARδ (custom polyclonal to mouse PPARδ) (*Fan et al., 2017*), SMRT (custom polyclonal to mouse SMRT) (*Barish et al., 2012*), NCoR (custom polyclonal to mouse NCoR) (*Barish et al., 2012*), HDAC3 (Santa Cruz) or H3K27ac (Active Motif). Antibody complexes were precipitated with IgG paramagnetic beads (ThermoFisher) for ChIP-seq or Protein A agarose beads (Millipore) for ChIP followed by qPCR. DNA was decrosslinked and purified using MinElute PCR purification columns (Qiagen). ChIP DNA was either assessed via qPCR and expressed as percent recovery of input chromatin or further processed into libraries for ChIP-seq. See *Supplementary file 1* for ChIP qPCR primers.

## ChIP sequencing

Sequencing libraries were generated from ChIP DNA using KAPA DNA Library Preparation kits (Kapa Biosystems) according to manufacturer's instructions. Libraries were assessed by Bioanalyzer

(Agilent) and qPCR-based quantification (Kapa Biosystems) and sequenced on an Illumina NextSeq 500 instrument using 75 bp single-end reads. Raw sequence reads were aligned to a reference genome (mm10) using Bowtie version 1.1.1 (*Langmead et al., 2009*) using '-m 1' and '–best' parameters to ensure reporting of uniquely mapped reads. Tag directories were generated using 'makeTagDirectory' using the -tbp 1 option to limit the number of reads starting at the same position to 1. ChIP-seq peaks were identified and analyzed using HOMER (*Heinz et al., 2010*). ChIP-seq peaks were identified in HOMER using the 'getDifferentialPeaksReplicates.pl' command, specifying '-style factor' to generate a high confidence set of peaks across triplicate samples. This command generates a peak list in three steps: first, it pools target tag directories to perform an initial peak identification against input; second, it quantifies raw reads of each target and input tag directory at the initial putative peaks; third, it calls DESeq2 to calculate enrichment values for each peak using the individual raw counts and returns only those peaks that pass two fold enrichment and FDR < 0.05. Peaks were annotated to nearest genes using 'annotatePeaks.pl.'

To characterize enriched motifs near BCL6-binding sites, we used HOMER's 'findMotifsGenome. pl' command to scan 50 bp windows surrounding BCL6 peaks, including the -mask option compared to random whole genome sequences. Motif densities were then quantified using HOMER's 'annotatePeaks.pl' using known motifs for PPARE, RXR, LXRE, and FXR; the BCL6 motif displayed in the density plots was identified with HOMER's *de novo* motif discovery tool using 200 bp scanning windows surrounding BCL6 peaks. Motif finding near SMRT, NCoR, and HDAC3 peaks used a 200 bp scanning window; the top 20 motifs by p-value were included in the heatmap. Enriched motifs were identified in all peaks for each factor compared to random whole genome sequences. Peak sets were then each compared to BCL6 to identify enriched motifs at shared sites against DNA sequences from unique sites. Conversely, for each cofactor, enriched motifs were identified at unique sites compared against DNA sequences from peaks shared with BCL6.

To generate the tag density scatter plots and histograms, tags were quantified using HOMER's 'annotatePeaks.pl' command, with either '-size 400' option or '-size 2000 -hist 25' options, for scatter plots and histograms, respectively. HOMER's 'mergePeaks' was used to compare different peak sets, defining overlapping peaks as those with a maximum distance between peak centers of 200 bp. BigWig browser tracks were generated using HOMER's 'makeMultiWigHub.pl' program and then viewed on the UCSC Genome Browser (*Kent WJ et al., 2002*). Gene ontologies for ChIP-seq data were generated using GREAT (*McLean et al., 2010*) by annotating ChIP-seq peaks to the single nearest gene.

To calculate distance between BCL6 and PPARα peaks, the distance between each BCL6 peak and the nearest PPARα peak was calculated using HOMER's 'annotatePeaks.pl' command and the '-pdist' option. Distances < 200 bp were plotted in a histogram where each bin represents increasing increments of 10 bp.

The four-way Venn was generated using Intervene (*Khan and Mathelier, 2017*).

## RNA sequencing and analysis

Liver samples (<30 mg) were stored in 1 mL of RNAlater Stablization Solution (Ambion) at −80° immediately following harvest. To isolate total RNA, tissues were homogenized in 1 mL buffer RLT (Qiagen) using the Mo Bio Powerlyzer. RNA was isolated and purified using RNeasy columns according to the manufacturer's protocol (Qiagen). RNA quality was assessed using a Bioanalyzer (Agilent) to ensure a RIN score greater than 7.0.

Sequencing libraries were constructed from purified RNA using the KAPA Stranded RNA-seq Kit with RiboErase (HMR) according to the manufacturer's instructions. Libraries were quantified using both a Bioanalyzer (Agilent) and qPCR-based quantification (Kapa Biosystems) and sequenced on an Illumina NextSeq 500 instrument using 75 bp single-end reads.

RNA raw sequence reads were aligned to a reference genome (mm10) using STAR version 2.4.0 hr (*Dobin et al., 2013*). Aligned reads included only unique mappers and those with fewer than four mismatches. Gene expression at exons was quantified using HOMER (*Heinz et al., 2010*). Differentially expressed RNAs were then normalized and identified using DESeq2 version 1.14.1 (*Love et al., 2014*) with an adjusted FDR < 0.05. Direct comparisons were made between *Bcl6*^fl/fl fed and fasted animals, as well as between fed *Bcl6*^fl/fl and *Bcl6*^LKO animals to generate lists of differentially expressed genes.

To compare the BCL6 cistrome and transcriptome, BCL6 ChIP-seq peaks were annotated to the nearest transcription start site using 'annotatePeaks.pl' in HOMER and then grouped based on the liver gene expression change of the annotated gene in $Bcl6^{LKO}$ mice compared to $Bcl6^{fl/fl}$ controls. Peaks were grouped as 'repressed,' 'activated,' or 'unchanged' BCL6 peaks if gene expression was higher, lower, or unaffected in $Bcl6^{LKO}$, respectively.

To determine 'rescued' gene expression in $Ppara^{-/-};Bcl6^{LKO}$ animals, impaired gene expression in $Ppara^{-/-}$ animals was first defined. To do this, we first identified genes that were significantly changed (adjusted p-value<0.05) with fasting in $Bcl6^{fl/fl}$ control animals. Then, among genes normally upregulated with fasting, we identified genes that were significantly less expressed (adjusted p-value<0.05) in fasted $Ppara^{-/-}$ compared to fasted control mice. Similarly, we identified genes normally downregulated with fasting that were significantly more expressed in fasted $Ppara^{-/-}$ compared to control fasted mice. Collectively, these significantly different up- and downregulated genes represent dysregulated genes in fasted $Ppara^{-/-}$ mice. To then determine rescued gene expression in $Ppara^{-/-};Bcl6^{LKO}$ animals, we identified dysregulated $Ppara^{-/-}$ genes that demonstrated no significant difference in gene expression between control fasted mice and $Ppara^{-/-};Bcl6^{LKO}$ fasted mice, indicating a restoration of fasting gene expression to control levels. We also identified partially rescued gene expression. Upregulated fasting genes with significantly lower expression in fasting $Ppara^{-/-}$ animals were considered partially rescued if fasting $Ppara^{-/-};Bcl6^{LKO}$ gene expression was significantly higher than $Ppara^{-/-}$ but also still significantly lower than control gene expression. Similarly, downregulated fasting genes with significantly higher expression in fasting $Ppara^{-/-}$ animals were considered partially rescued if fasting $Ppara^{-/-};Bcl6^{LKO}$ gene expression was significantly lower than $Ppara^{-/-}$ but also still significantly higher than control gene expression.

Gene ontology analysis was performed on differential genes with an adjusted p-value<0.05 and a log fold change greater than one using Metascape (*Tripathi et al., 2015*). Enrichment analysis included terms from Reactome Gene Sets, GO Biological Processes and KEGG Pathways.

RPKM values were generated using HOMER (*Heinz et al., 2010*) and displayed as heatmaps using Morpheus (*Gould, 2019*). The distance matrix heatmap was generated using the 'dist' function in R version 3.4.3 (*R Development Core Team, 2017*) and plotted with the heatmap.2 function and the 'RColorBrewer' package (*Neuwirth, 2014*).

To generate the heatmap of H3K27ac and PPARδ enrichment at BCL6-PPARα peaks near rescued genes, relevant BCL6-PPARα-bound regions were identified by annotating peaks to the nearest transcription start site using HOMER. H3K27ac ChIP-seq enrichments in fasted control, $Ppara^{-/-}$ and $Ppara^{-/-};Bcl6^{LKO}$ samples were then quantified at these peaks using 'annotatePeaks.pl' and the '-size 6000 -hist 25 -ghist' options; PPARδ ChIP-seq enrichment was quantified using '-size 3000.' The signal averages across biological replicates were plotted as a heatmap using Morpheus.

All UCSC genome browser tracks represent combined tag directories across replicates.

Genes differentially expressed with fasting were classified based on their association with regulatory regions. First, shared BCL6-PPARα ChIP-seq peaks were annotated to the single nearest gene using HOMER. Using these annotations, genes differentially expressed with fasting (adjusted p-value<0.05) were then grouped based on presence or absence of a nearby shared BCL6-PPARα annotated peak. Of those differential fasting genes without an annotated shared BCL6-PPARα peak, genes were further grouped based on presence of annotated BCL6 unique and PPARα unique ChIP-seq peaks. Some genes had both BCL6 unique and PPARα unique peaks, but these were non-overlapping. Other genes had neither BCL6- nor PPARα-annotated peaks nearby. The frequency of BCL6-PPARα, PPARα only, or BCL6 only peaks per fasting gene was also calculated.

## qPCR analysis

Frozen liver tissues were homogenized in Trizol (Ambion) using a Mo Bio Powerlyzer. Chloroform was added at 200 µL to 1 mL homogenates in Trizol. The clear aqueous phase was extracted after centrifugation. RNA was then isolated with a RNeasy kit (Qiagen), according to manufacturer's protocol. cDNA was synthesized with 600–1000 ng of RNA using the iScript cDNA Synthesis Kit (Bio-Rad). Gene expression was then assessed via qPCR using iTaq Universal SYBR Green Supermix (BioRad). Gene expression was normalized to the housekeeping gene, *36b4*. See *Supplementary file 2* for primer sequences.

## Histology

For the hematoxylin and eosin (H & E) staining, liver tissues were fixed in 10% formalin overnight and then moved to 70% EtOH. Fixed tissues were paraffin embedded, cut, and stained by the Northwestern University Research Histology and Phenotyping Laboratory which is supported by NCI P30-CA060553 awarded to the Robert H. Lurie Comprehensive Cancer Center. For oil red O staining, liver samples frozen in OCT were cut to 5–7 µm with a Leica cryostat, mounted onto slides, stained with oil red O, and counterstained with hematoxylin.

## Lipid and metabolite measurements

We measured serum triglycerides (Infinity Thermo Fisher) and ketone bodies (Cayman Chemical) using commercial kits. To measure tissue triglycerides, we extracted lipids using a modified version of the Folch Method (*Folch et al., 1957*). In brief, tissues were homogenized in 1 mL of methanol using the Mo Bio Powerlyzer. Homogenates were transferred to glass tubes and incubated several hours in 1:2 methanol:chloroform after briefly vortexing. 0.9% NaCl was added to homogenates overnight to separate the chloroform lipid-containing layer from the methanol layer. The next day, the methanol and any floating tissue was aspirated. The remaining chloroform layer was dried under nitrogen gas. Lipid was resuspended in 2-propanol and quantified using the Infinity Thermo Fisher triglyceride kit. Quantified lipid was normalized to tissue weight. Serum insulin was measured via ELISA (Crystal Chem) and serum glucose was measured using a colorimetric assay (BioVision).

## Hepatic triglyceride secretion

We fasted mice for 4 hr and then injected mice intraperitoneally with a 7.5% poloxamer solution in PBS at a dose of 1 mg/g body weight. Tail-vein blood samples were collected over time using capillary Microvettes (Sarstedt).

## *De novo* lipogenesis

The rate of hepatic lipogenesis was determined via incorporation of $^2$H into newly made TG-bound fatty acids, as described elsewhere (*Bederman et al., 2006*). Briefly, mice were injected i.p. with 0.7 mL of $^2$H-labeled saline (9 g of NaCl in 1 L of 99.9% $^2$H$_2$O). For the next 24 hr, mice were maintained on 6% $^2$H-labeled drinking water and then harvested. Terminal serum and liver tissue samples were collected and flash frozen. Sample processing and GC/MS analysis was performed as described previously (*Bederman et al., 2012*).

## Fatty acid oxidation

We determined rates of fatty acid oxidation in liver homogenates by measuring oxidation of $^{14}$C palmitate (*Hirschey and Verdin, 2010*). Briefly, tissue was dounce homogenized in sucrose/Tris/EDTA buffer and incubated for 30 or 60 min in a reaction mixture containing 0.4 uCi $^{14}$C palmitate. After reacting with the labeled palmitate, mixtures were transferred to tubes containing 1M perchloric acid with Whatman paper discs soaked in 1M NaOH in the lids. Scintillation counting was used to measure $^{14}$C in the acid-soluble fraction and in disc-trapped $CO_2$, representing partially and fully oxidized radiolabeled palmitate, respectively. Fatty acid oxidation rates were then expressed as amount of substrate oxidized per tissue weight per minute.

## *In vivo* lipid uptake

Mice were injected with BODIPY-C16 (Life Technologies) to assess lipid uptake, as described elsewhere (*Wilson et al., 2016*). BODIPY-C16 was resuspended in dimethylsulfoxide at 10 mM. Then, a 0.1 µg/µL working stock was made in 0.25% fatty-acid-free BSA (Sigma-Aldrich) solution in PBS. Mice were fasted for 4 hr and then injected intraperitoneally with BODIPY-C16 at 0.5 µg/g of body weight. After 5 hr, tissues were collected and flash frozen. 80–120 mg of liver tissues were dounce homogenized in RIPA buffer. 25 µL volumes of cleared tissue homogenates were diluted 1:4 in PBS and analyzed using a fluorescent plate reader (Ex 485 nm, Em 515 nm). Saline-injected mouse liver homogenates were used to control for background fluorescence. Tissue fluorescence was normalized to tissue weight.

## Insulin signaling assay

Mice were placed on 5 weeks of high-fat diet. After a 5 hr fast, mice were injected intraperitoneally with 1 U/kg recombinant insulin. Ten minutes later, mice were sacrificed and their tissues were harvested.

## Western blotting

Frozen liver tissues were dounce homogenized in RIPA buffer. After incubating on ice for 10 min, homogenates were centrifuged at full speed for 15 min at 4°; supernatant was then collected and stored at −80°. Protein was quantified with a BCA assay (Thermo Scientific) and 2 µg/µL lysates were boiled for 5 min in 5x loading buffer. Denatured protein lysates were loaded in precast polyacrylamide gels (BioRad) and transferred to PVDF membranes (BioRad). Membranes were blocked with 5% milk in PBST and probed with primary antibodies for BCL6 (Santa Cruz, D-8) at 1:200, PPARα (Santa Cruz, H-98) 1:500, pAKT (Cell Signaling, 4060) 1:1000, panAKT (Cell Signaling, 4691) 1:1000 or β-actin (Sigma, A1978) 1:1000 overnight at 4°. Secondary antibodies were added for 1 hr at room temperature (Jackson ImmunoResearch). Protein was then visualized using ECL (ThermoScientific). MemCode Reversible Stain was used to visualize total protein (Thermo Fisher Scientific). Protein densitometry was quantified using ImageJ 1.51 s (*Schneider et al., 2012*).

## Accession numbers

All RNA-seq and ChIP-seq data are deposited in GEO SuperSeries accession #GSE118789.

## Acknowledgements

We thank Joe Bass, Liming Pei, and Debabrata Chakravarti for helpful advice and discussion. We thank Northwestern University's Comprehensive Metabolic Core and Mouse Histology and Phenotyping Laboratory (supported by NCI P30-CA060553 awarded to the Robert H Lurie Comprehensive Cancer Center) for services. This work was funded by NIH grants R01DK108987 (GB) and K08HL092298 (GB), American Diabetes Association Award 1–17-IBS-137 (GB), and NIH T32 GM008061 (MS).

## Additional information

### Funding

| Funder | Grant reference number | Author |
|---|---|---|
| National Institutes of Health | R01DK108987 | Grant D Barish |
| American Diabetes Association | 1-17-IBS-137 | Grant D Barish |
| National Institutes of Health | K08HL092298 | Grant D Barish |
| National Institutes of Health | T32GM008061 | Meredith A Sommars |

The funders had no role in study design, data collection and interpretation, or the decision to submit the work for publication.

### Author contributions

Meredith A Sommars, Conceptualization, Data curation, Formal analysis, Funding acquisition, Validation, Investigation, Visualization, Methodology, Writing—original draft, Writing—review and editing; Krithika Ramachandran, Christopher R Futtner, Conceptualization, Data curation, Formal analysis, Investigation, Methodology, Writing—original draft, Writing—review and editing; Madhavi D Senagolage, Derrik M Germain, Data curation, Formal analysis, Investigation, Methodology, Writing—original draft, Writing—review and editing; Amanda L Allred, Yasuhiro Omura, Conceptualization, Data curation, Formal analysis, Investigation, Writing—original draft, Writing—review and editing; Ilya R Bederman, Conceptualization, Resources, Data curation, Formal analysis, Investigation, Writing—original draft, Writing—review and editing; Grant D Barish, Conceptualization, Resources,

Formal analysis, Supervision, Funding acquisition, Investigation, Writing—original draft, Writing—review and editing

### Author ORCIDs
Ilya R Bederman http://orcid.org/0000-0002-1909-5746
Grant D Barish http://orcid.org/0000-0001-8753-0546

### Ethics
Animal experimentation: All animal care and use procedures were conducted in accordance with regulations of the Institutional Animal Care and Use Committee at Northwestern University, protocol IS00004929.

### Decision letter and Author response
Decision letter https://doi.org/10.7554/eLife.43922.029
Author response https://doi.org/10.7554/eLife.43922.030

## Additional files

### Supplementary files
• Supplementary file 1. Quantitative PCR primers for chromatin immunoprecipitation.
DOI: https://doi.org/10.7554/eLife.43922.019

• Supplementary file 2. Quantitative PCR primers for gene expression.
DOI: https://doi.org/10.7554/eLife.43922.020

• Transparent reporting form
DOI: https://doi.org/10.7554/eLife.43922.021

### Data availability
Sequencing data have been deposited in GEO under SuperSeries accession #GSE118789.

The following dataset was generated:

| Author(s) | Year | Dataset title | Dataset URL | Database and Identifier |
|---|---|---|---|---|
| Barish GD | 2018 | BCL6 de-repression induces the fasting transcriptome and protects from steatosis | https://www.ncbi.nlm.nih.gov/geo/query/acc.cgi?acc=GSE118789 | NCBI Gene Expression Omnibus, GSE118789 |

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
