## [Decision Letter]

Thank you for submitting your article "Dynamic repression by BCL6 controls the genome-wide liver response to fasting and steatosis" for consideration by *eLife*. Your article has been reviewed by three peer reviewers, one of whom is a member of our Board of Reviewing Editors, and the evaluation has been overseen by Mark McCarthy as the Senior Editor. The reviewers have opted to remain anonymous.

The reviewers have discussed the reviews with one another and the Reviewing Editor has drafted this decision to help you prepare a revised submission.

Summary:

Sommars and colleagues focused on the role of the transcriptional repressor BCL6 in the regulation of hepatic lipid metabolism. The authors report that BCL6 acts as an antagonist of the PPARalpha transcriptional program, and inhibits the transcriptional signature induced in liver by fasting. They demonstrate that BCL6 and PPARalpha bind independently to many regulatory regions. The authors also show that BCL6 liver knockout mice activate a transcriptional program resembling the fasting-induced response, including changes in genes linked to increased lipid β-oxidation and ketogenesis. Accordingly, hepatic deletion of BCL6 was shown to protect mice from HFD-induced steatosis. The authors also demonstrate that BCL6 deletion in liver of PPARα knockout mice reverses the impaired response to fasting and ameliorates fasting-induced hepatic lipid accumulation. Overall, the reviewers found the study to be well done and the findings to be of broad interest. The reviewers also identified several points that should be further explored in order to strengthen the conclusions of the manuscript.

Essential revisions:

1) Bcl6 ablation rescues fasting gene expression defects in PPARa -/- mice. In this setting, what is driving the fasting transcriptional program? Do the authors have any hypotheses about this that can be tested experimentally?

2) Since the genome-wide DNA binding of BCL6 and PPARa are mutually independent and in Discussion the authors propose that BCL6 opposition to PPARa occurs via proximate binding at independent cis-regulatory elements, it would be helpful to plot the genomic binding sites of BCL6 surrounding the nearby PPARa sites or the other way around. Do the BCL6 and PPARa sites overlap? Are these factors competing with one another physically?

3) In Figure 4A, are there any unique feature of genes that cannot be restored expression upon fasting in Ppara-/-; Bcl6 LKO mice?

4) If the experiment is feasible in a two-month time frame, it would be interesting to know What happens if you overexpress BCL6 in a fasted mouse? Does this cause a reduction of fatty acid oxidation? Such an experiment would enhance the paper, but it is not considered essential for publication.

5) In Figure 5F and 5G, the attenuation of steatosis in 19-week of HFD-fed Bcl6 LKO mice is a very positive finding. Have the authors checked to see if there is any effect on the HFD-induced glucose intolerance and insulin resistance of these mice?

6) It is known that BCL6 acts as a transcriptional repressor by interacting with many co-repressors, including SMRT, NCoR and HDACs. It is known that NCoR1 and SMRT are involved in regulation of lipid metabolism in liver, as their ablation causes hepatic steatosis (Shimizu et al., 2015). Do the authors have any insight into the molecular mechanism by which BCL6 represses transcription in liver?

---

## [Author Response]

Essential revisions:1) Bcl6 ablation rescues fasting gene expression defects in PPARa -/- mice. In this setting, what is driving the fasting transcriptional program? Do the authors have any hypotheses about this that can be tested experimentally?

We thank the reviewers for this important question. Rescued genes in *Ppara^-/-^;Bcl6^LKO^*fasted livers are significantly more likely to be associated with regulatory regions containing BCL6-PPARa binding sites. Accordingly, to address how fasting gene expression defects can be rescued, we have focused on understanding epigenomic regulation at these regions. We mapped acetylated H3K27 (a marker for enhancer activity) in *Ppara^-/-^* and *Ppara^-/-^;Bcl6^LKO^*fasted livers. We found that H3K27ac was increased at enhancers associated with upregulated rescued genes, but it was not changed at non-rescued genes. (Figure 5C and Figure 5—figure supplement 2B). These results implied that reduced histone deacetylase (HDAC) activity, increased histone acetyltransferase (HAT) activity, or possibly both changes at BCL6-PPARα regulatory regions could underlie the rescued fasting expression observed in *Ppara^-/^;Bcl6^LKO^* animals.

BCL6 functions as a repressor and can complex with various HDACs, including HDAC3 (as demonstrated in Figure 3C and D). We found that HDAC3 recruitment to shared BCL6-PPARα sites along rescued genes was significantly diminished with loss of *Bcl6* (Figure 5D). Of note, although BCL6 levels are reduced with fasting, BCL6 still retains ~50% of its genome-wide occupancy under these conditions (Figure 1H and Figure 1—figure supplement 2A). Compared to fasting *Ppara^-/-^* mice, complete *Bcl6* ablation in *Ppara^-/-^;Bcl6^LKO^*mice is expected to cause a further decline in HDAC3 occupancy on BCL6-PPARα-controlled fasting enhancers. Thus, loss of BCL6 repression and associated HDACs is likely to increase fasting enhancer activity and the expression of associated genes.

In addition, we tested whether loss of both *Bcl6* and *Ppar*a could impact the recruitment of activating transcription factor complexes to BCL6 or PPARα binding sites. Specifically, we examined STAT5 and PPARδ, which have each been associated with p300/CBP (Jin et al., 2011; Pfitzner, Jahne, Wissler, Stoecklin, and Groner, 1998). It has previously been suggested that BCL6 and STAT5 can compete for binding at growth-hormone regulated targets in liver (Zhang, Laz, and Waxman, 2012). To test if genetic loss of *Bcl6* promoted STAT5 binding at rescued genes, we analyzed STAT5 occupancy genome-wide in fasted

*Ppara*^-/-^ and *Ppara^-/-^;Bcl6^LKO^*livers. However, we did not find enhanced recruitment of STAT5 to BCL6-PPARα shared enhancers near rescued genes (Author response image 1).

**Author response image 1. respfig1:** Loss of BCL6 does not enhance STAT5 recruitment to shared BCL6-PPARα enhancers. (**A**) STAT5 ChIP-seq tag density in *Ppara*^-/-^ and *Ppara^-/-^;Bcl6^LKO^*livers at BCL6-PPARα peaks near rescued upregulated and downregulated fasting genes. ChIP was performed in biological triplicate. (**B**) Heatmap of STAT5 ChIP-seq in fasted control *Bcl6^fl/fl^*, *Ppara*^-/-^, and *Ppara^-/-^;Bcl6*^*LKO*^mice at BCL6-PPARα shared peaks that annotate to rescued *Ppara*^-/-^dysregulated genes in *Ppara^-/-^;Bcl6^LKO^*mice.

Since PPARα and PPARδ compete for binding at shared sites (Figure 2 —figure supplement 1C), we tested whether enhanced recruitment of PPARδ to shared BCL6-PPARα regions might drive expression of fasting genes in the combined absence of *Ppar*a and *Bcl6*. Notably, we observed that *Ppar*d expression is higher in fasted *Ppara^-/^;Bcl6^LKO^*mice compared to *Ppara^-/-^* mice (Figure 5E), and BCL6 binds along the *Ppard* gene (Figure 5—figure supplement 2C). These results indicated that BCL6 directly represses *Ppard* expression. We performed ChIP-sequencing and found that PPARδ binding was enhanced at shared BCL6-PPARα sites near rescued genes in fasting *Ppara^-/-^;Bcl6^LKO^*mice compared to wild type control or *Ppara^-/-^* livers (Figure 5C and F), particularly at fasting upregulated genes. Thus, loss of *Bcl6* facilitates compensation for PPARα deficiency via PPARδ.

In summary, we believe that the rescued fasting expression observed in *Ppara^-/-^;Bcl6^LKO^*mice is achieved both through reduced BCL6-mediated repression and enhanced activation by PPARδ at BCL6-PPARα controlled fasting enhancers.

2) Since the genome-wide DNA binding of BCL6 and PPARa are mutually independent and in Discussion the authors propose that BCL6 opposition to PPARa occurs via proximate binding at independent cis-regulatory elements, it would be helpful to plot the genomic binding sites of BCL6 surrounding the nearby PPARa sites or the other way around. Do the BCL6 and PPARa sites overlap? Are these factors competing with one another physically?

Thank you for this suggestion. We have now included a histogram plotting the distance between BCL6 peaks and the nearest PPARα peak center (Figure 1—figure supplement 2). We find that even within a 200-bp window, >96% of BCL6 and PPARα peak centers are within 100 bp of each other. Despite their close proximity, we find no evidence that these factors are competing with each other physically. Their consensus sites are distinct (Figure 1B), and genetic ablation of either factor does not alter chromatin recruitment of the other factor (Figure 2A and Figure 1—figure supplement D-E).

3) In Figure 4A, are there any unique feature of genes that cannot be restored expression upon fasting in Ppara-/-; Bcl6 LKO mice?

The reviewers raise a very interesting question. It should be noted that in our revised manuscript, Figure 4A has now become Figure 5A. We have found several epigenomic features that typify the regulatory regions associated with genes that cannot be restored compared to those that can rescued.Unrestorable features include (*i*) static H3K27 acetylation, (*ii*) relatively modest recruitment of PPARδ, (*iii*) stronger PPARα binding, (*iv*) lack of BCL6 binding sites, and (*v*) distinct motif enrichments. We detail these features below.

i) As discussed in Question #1, we found that enhancer activity correlates to transcriptional rescue. H3K27ac marks were relatively unchanged near non-rescued genes (Figure 5—figure supplement 2B). In contrast, H3K27ac was increased near rescued upregulated genes or reduced at rescued downregulated genes (Figure 5C).

ii) A potential contributor to these differences in H3K27 acetylation is PPARδ recruitment. Rescued fasting genes are associated with increased PPARδ binding in *Ppara^-/-^;Bcl6^LKO^* mice (Figure 5F). At non-rescued genes, we also observe PPARδ recruitment to nearby BCL6-PPARα binding sites in *Ppara-/-;Bcl6^LKO^*animals (Author response image 2), but the magnitude of the increase in PPARδ binding is significantly less than at rescued genes (Author response image 2).

iii) We observed that PPARα binding strength influences whether a gene can be rescued. We have analyzed PPARα binding strength on binding sites of rescued versus nonrescued genes based on our ChIP-sequencing data in wild type mice (Author response image 2). PPARα binding is stronger at its peaks near non-rescued genes compared to rescued genes, particularly under fasting conditions. Thus, *Bcl6* ablation may be insufficient to upregulate genes with particularly strong PPARα-dependence.

iv) Although BCL6 binding strength is not different among rescued versus non-rescued genes (Author response image 2), genes that are restored in *Ppara^-/-^*mice by *Bcl6* ablation are significantly more likely to have a nearby BCL6 binding site than to lack such sites. This indicates an important role for relief of BCL6-mediated active repression to rescue defective expression in *Ppara^-/-^* mice.

v) Additionally, differential recruitment of other transcription factors to nearby enhancers may influence which genes are rescued with loss of *Bcl6*. We identified all active enhancers that map to rescued or non-rescued genes in *Ppara^-/-^;Bcl6^LKO^* mice by annotating H3K27ac ChIP-seq peaks to their nearest transcription start sites and performed motif analysis of these distinct sets of regulatory regions. Using non-rescued enhancers as background, we identified enriched motifs for homeobox family transcription factors (NKX6.1, PITX1), heat shock factor, ZBTB18, and FOXO near rescued genes (Author response image 2). In contrast, using rescued enhancers as background, we identified motifs for MYB, NR5A2 (LRH-1), E2F1, ZNF675, and OCT4 in enhancers near non-rescued genes.

**Author response image 2. respfig2:** Rescued genes are enriched for nearby PPARδ binding and motifs for distinct transcription factors compared to non-rescued genes. (**A**) PPARδ ChIP-seq tag density in *Ppara^-/-^* and *Ppara^-/-^;Bcl6^LKO^*livers at shared BCL6-PPARα peaks near up- and downregulated non-rescued fasting genes. (**B**) Log_2_ ratio of PPARδ ChIP-seq tag density in *Ppara^-/-^;Bcl6^LKO^*over *Ppara^-/-^* livers at BCL6-PPARα peaks near rescued and non-rescued fasting genes. (**C**) PPARα tag density (left) and BCL6 tag density (right) in control fed and fasted livers at respective peaks near rescued and non-rescued genes. (**D**) Top 10 enriched motifs at enhancers identified by H3K27ac ChIP-seq in fasted *Ppara^-/-^;Bcl6^LKO^*livers near rescued (left) and nonrescued (right) genes. Motifs near rescued genes were identified against DNA sequences in non-rescued enhancers; motifs near non-rescued genes were identified against DNA sequences in rescued enhancers. Box plots display interquartile range (box), median (horizontal line), mean (black ‘+’), and min to max (whiskers). Mann-Whitney test was used to compare tags between groups. p < 0.05, p < 0.01, p < 0.001.

4) If the experiment is feasible in a two-month time frame, it would be interesting to know What happens if you overexpress BCL6 in a fasted mouse? Does this cause a reduction of fatty acid oxidation? Such an experiment would enhance the paper, but it is not considered essential for publication.

To explore the effects of *Bcl6* gain-of-function, we have very recently developed a *Rosa* locus transgenic model to constitutively overexpress *Bcl6* in mouse liver. In fasting mice, transgenic animals (*Bcl6*^LTG^ mice) express 40-fold higher levels of *Bcl6* RNA, and protein levels are approximately 35-fold higher than in wild type controls (Author response image 3). We find that many lipid metabolism genes which were upregulated by genetic ablation of *Bcl6* are reciprocally downregulated by overexpressing *Bcl6* (Author response image 3). However, when placed on a 48-hour fast, Bcl6^LTG^ mice did not exhibit a difference in liver triglyceride accumulation (Author response image 3). Given the promising changes in gene expression, we hypothesize that a more extended lipid challenge may reveal a phenotype. We plan to perform high fat diet studies in the future to test whether *Bcl6*overexpression may exacerbate steatosis in that context.

**Author response image 3. respfig3:** Overexpression of BCL6 downregulates fatty acid oxidation genes. (**A**) qPCR analysis of *Bcl6* in control and *Bcl6^LTG^* livers. N = 3-7 per group. (**B**) Western blot of BCL6 protein in control and *Bcl6^LTG^* livers. Densitometric quantification is shown normalized to actin (right). (**C**) qPCR analysis of fatty acid oxidation and ketogenesis genes after a 48-hour fast. N = 6-7 per group. (**D**) Hepatic triglyceride content after a 48-hour fast. N = 67 per group. A two-tailed Student’s t-test assuming equal variance was used to compare mean values between groups. Data are represented as mean ± SEM. p < 0.05, p < 0.01, p < 0.001.

5) In Figure 5F and 5G, the attenuation of steatosis in 19-week of HFD-fed Bcl6 LKO mice is a very positive finding. Have the authors checked to see if there is any effect on the HFD-induced glucose intolerance and insulin resistance of these mice?

We have tested fasting serum glucose and insulin levels after 17 weeks of high fat diet. We found that glucose was significantly lower and insulin trended lower in mice lacking *Bcl6* (Figure 6—figure supplement 1A and B). To further explore whether protection from hepatic triglyceride accumulation improved insulin sensitivity, we placed mice on a shorter term of high fat diet (5 weeks) and then measured insulin signaling *in vivo*. We injected mice with insulin (1 unit/kg), harvested their livers after ten minutes, and then performed western blots to measure total and phosphorylated-AKT (p-AKT). *Bcl6^LKO^*mice exhibited higher levels of p-AKT and ratios of p-AKT to total AKT, suggesting improved hepatic insulin sensitivity (Figure 6—figure supplement 1C).

6) It is known that BCL6 acts as a transcriptional repressor by interacting with many co-repressors, including SMRT, NCoR and HDACs. It is known that NCoR1 and SMRT are involved in regulation of lipid metabolism in liver, as their ablation causes hepatic steatosis (Shimizu et al., 2015). Do the authors have any insight into the molecular mechanism by which BCL6 represses transcription in liver?

We thank the reviewers for this important question. It is intriguing that known BCL6 corepressors can be involved in very different lipid metabolic effects in liver. As the reviewers point out, disrupting NCoR interactions with nuclear receptors and globally disrupting SMRT interactions caused steatosis in the referenced manuscript. Likewise, deletion of liver *Hdac3* has been reported to cause steatosis (Sun et al., 2012). To better understand the mechanism of repression by BCL6 in liver, in our revised manuscript we have performed ChIP-sequencing of SMRT, NCoR, and HDAC3 in control and *Bcl6^LKO^* livers from *ad lib* fed mice using three biological replicates per condition. We produced robust datasets for analysis of these corepressors and their interactions with BCL6. We found extensive overlap between BCL6 and all three corepressors, with nearly 80% of BCL6 sites overlapping with SMRT, NCoR, or HDAC3, and nearly 40% of BCL6 sites overlapping with all three (Figure 3C). Moreover, in the genetic absence of *Bcl6,* we found that these cofactors exhibited significantly reduced recruitment to regions with BCL6 binding sites (Figure 3D), which was accompanied by increased H3K27 acetylation (Figure 3—figure supplement 2B). Together, these results provided evidence that liver BCL6 recruits SMRT/NCoR and HDAC3 to repress a subset of its *cis*-elements.

However, BCL6 binds to 3,728 sites independently of SMRT, NCoR, or HDAC3, and these corepressors bind to thousands of regions independently of BCL6 (Figure 3C). This was particularly apparent for NCoR, which exhibited over 35,000 binding sites in liver. These results suggest that BCL6 and SMRT/NCoR-HDAC3 can regulate independent transcriptional networks. Importantly, even with *Bcl6* ablation, residual corepressor binding of SMRT, NCoR, and HDAC3 is apparent on regions to which BCL6 is normally bound (Figure 3D). This implies that SMRT, NCoR, and HDAC3 may frequently engage multiple transcription factor complexes within a single regulatory region. Given that SMRT, NCoR, and HDAC3 can interact with many different transcription factors (Figure 3—figure supplement 2), it is perhaps not surprising that phenotypes for these corepressor knockouts are quite different than the phenotype observed here in *Bcl6*^LKO^ mice. In the future, we hope to further define BCL6 corepressor complexes in liver and the contribution of SMRT/NCoR-HDAC3 to BCL6-mediated lipid regulation.

Additionally, to gain insight into gene regulation by BCL6 and SMRT/NCoR-HDAC3, we compared the transcriptional effects of *Bcl6* deletion to *Smrt-NCoR* or *Hdac3* deletion by comparing the our BCL6 RNA-seq data to other published datasets (Shimizu et al., 2015; Sun et al., 2012). We found 799 genes differentially expressed (p-value < 0.05) with both *Bcl6* deletion and *Smrt*KO-*Ncor* RID double deletion (DKO)(Author response image 4). Among these, we identified four gene expression categories (up in *Bcl6^LKO^*/up in DKO; up in *Bcl6^LKO^*/down in DKO; down in *Bcl6^LKO^*/up in DKO; down in *Bcl6^LKO^*/down in DKO). Using these four categories, we performed a clustering analysis of enriched gene ontologies. Quite interestingly, we found that genes downregulated in *Bcl6^LKO^* mice and upregulated in DKO mice were enriched for functions in fatty acid biosynthesis (Author response image 4). In other words, the upregulated genes presumably responsible for the steatosis observed in DKO mice are actually downregulated in *Bcl6*^LKO^ animals. Yet, genes upregulated in both models are enriched for biological oxidations, which was consistent with the role we describe for BCL6 to repress genes involved in fatty acid oxidation. In a similar manner, we compared differentially expressed genes from liver *Bcl6* deletion to those from liver *Hdac3* deletion (*Hdac3^LKO^*). We found 638 genes differentially expressed (p-value < 0.05) in both *Bcl6^LKO^* and *Hdac3*^LKO^ mutant lines (Author response image 4). Genes up in both models were enriched for functions in biological oxidations (Author response image 4), but there was no signature for lipid synthesis or storage. However, among genes differentially expressed in *Hdac3^LKO^*but not in *Bcl6*^*LKO*^mice, we detected increased expression of lipid biosynthetic genes (Author response image 4). In summary, while some categories of SMRT/NCoR-regulated or HDAC3-regulated genes resemble BCL6-regulated genes, overall expression is not well correlated. This likely underlies the opposing phenotypes of liver SMRT/NCoR or HDAC3 models compared to *Bcl6^LKO^*mice.

**Author response image 4. respfig4:** BCL6 suppress fatty acid oxidation genes while SMRT/NCoR and HDAC3 suppress lipogenic genes. (**A**) Comparison of log fold change (LFC) at significant (p-value <0.05) differentially expressed genes in both *Bcl6^LKO^*and Ncor/Smrt double knockout (DKO) livers over control samples. (**B**) Pathway enrichment analysis for genes differentially expressed in *Bcl6^LKO^* and DKO livers based on direction of change. (**C**) Comparison of log fold change (LFC) at significant (p-value <0.05) differentially expressed genes in both *Bcl6^LKO^*and *Hdac3* knockout (*Hdac3*^LKO^) livers over control samples. (**D**) Pathway enrichment analysis for genes differentially expressed in *Bcl6^LKO^* and *Hdac3^LKO^* livers based on direction of change. (**E**) Pathway enrichment analysis for genes differentially expressed only in *Hdac3^LKO^* livers over control (left) and sub-pathway analysis of ‘monocarboxylic acid metabolic process’ (right).